# MoE-GS: Mixture of Experts for Dynamic Gaussian Splatting

**In-Hwan Jin**[1][*]  **Hyeongju Mun**[1][*]  **Joonsoo Kim**[2]  **Kugjin Yun**[2]  **Kyeongbo Kong**[1][†]

[1]Pusan National University, {ihjin, 201924128, kbkong}@pusan.ac.kr
[2]Electronics and Telecommunications Research Institute, {joonsookim, kjyun}@etri.re.kr

## Abstract

Recent advances in dynamic scene reconstruction have significantly benefited from 3D Gaussian Splatting, yet existing methods show inconsistent performance across diverse scenes, indicating no single approach effectively handles all dynamic challenges. To overcome these limitations, we propose Mixture of Experts for Dynamic Gaussian Splatting (MoE-GS), a unified framework integrating multiple specialized experts via a novel Volume-aware Pixel Router. Unlike sparsity-oriented MoE architectures in large language models, MoE-GS is designed to improve dynamic novel view synthesis quality by combining heterogeneous deformation priors, rather than to reduce training or inference-time FLOPs. Our router adaptively blends expert outputs by projecting volumetric Gaussian-level weights into pixel space through differentiable weight splatting, ensuring spatially and temporally coherent results. Although MoE-GS improves rendering quality, the increased model capacity and reduced FPS are inherent to the MoE architecture. To mitigate this, we explore two complementary directions: (1) single-pass multi-expert rendering and gate-aware Gaussian pruning, which improve efficiency within the MoE framework, and (2) a distillation strategy that transfers MoE performance to individual experts, enabling lightweight deployment without architectural changes. To the best of our knowledge, MoE-GS is the first approach incorporating Mixture-of-Experts techniques into dynamic Gaussian splatting. Extensive experiments on the N3V and Technicolor datasets demonstrate that MoE-GS consistently outperforms state-of-the-art methods with improved efficiency. Video demonstrations are available at `cvsp-lab.github.io/MoE-GS`.

## 1 Introduction

Realistically modeling dynamic scenes from real-world data is a fundamental challenge for training future AGI models, creating immersive content for spatial computing, and enabling embodied agents to effectively perceive and interact with their environments. Recent advances in novel view synthesis, particularly Neural Radiance Fields (NeRF) (Mildenhall et al., 2021), have significantly improved the quality of static scene reconstruction. However, NeRF's implicit representation and intensive ray-tracing introduce substantial computational overhead, limiting real-time applicability. To address these limitations, explicit representations such as 3D Gaussian Splatting (3DGS) (Kerbl et al., 2023) have emerged, achieving real-time rendering without compromising visual fidelity.

Recent efforts have extended Gaussian-based representations to dynamic scenes, introducing diverse approaches such as MLP-based deformation networks (Wu et al., 2024; Bae et al., 2024), polynomial-based motion models (Li et al., 2024), and interpolation-based methods (Lee et al., 2024). Although these methods achieve promising results in handling specific deformation types, our empirical analysis (Fig. 1) reveals that no single approach consistently generalizes across varied real-world dynamic scenarios. Specifically, our analysis identifies three key limitations:

- **Scene-level variations**: Different reconstruction methods exhibit significant variability in performance across scenes, indicating that each model has a restricted range of optimal applicability (Fig. 1a). Such scene-specific performance variations underscore the necessity for adaptive model selection.

- **Spatial-level inconsistencies**: Within same scene, reconstruction quality varies across different spatial regions when processed by distinct methods (Fig. 1b). This spatial variability demonstrates that no single model consistently excels across all regions of a given scene.

---

[*]Equal contribution.
[†]Corresponding author.

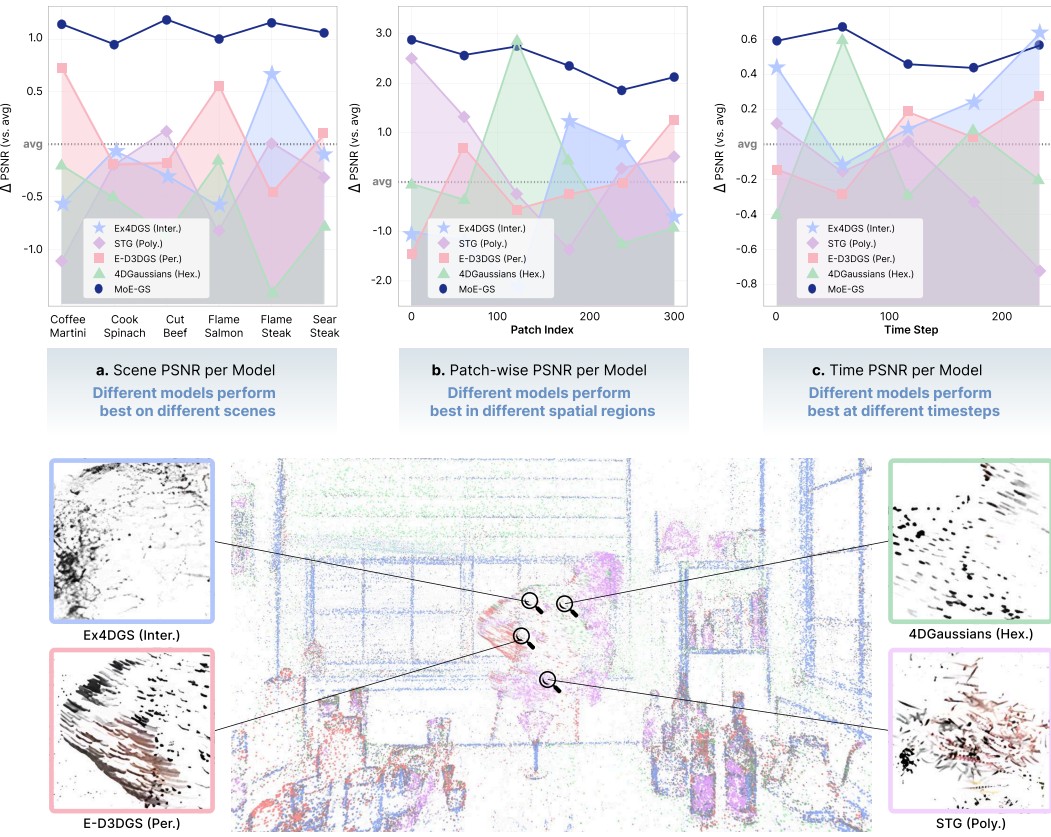

Figure 1: Limitations of existing dynamic Gaussian splatting methods. (a) Scene-level: No single method consistently dominates across scenes. (b) Spatial-level: Different spatial regions favor different deformation models. (c) Temporal-level: The best-performing method changes over time within the same scene. We also visualize representative motion trajectories of four experts—4DGaussians (Green), STG (Purple), E-D3DGS (Pink), and Ex4DGS (black)—to illustrate their distinct motion behaviors. Additional video results are provided on the project page.

- **Temporal fluctuations**: Within a video sequence, the best-performing method changes dynamically from frame to frame, reflecting the inherent temporal complexity of real-world dynamics (Fig. 1c). Such temporal fluctuations highlight the inability of individual models to consistently capture complex temporal patterns.

These variations arise from the heterogeneous inductive biases of existing dynamic GS methods. Each deformation model is naturally suited to a particular motion regime: HexPlane-based canonical deformation produces smooth, highly regularized trajectories that work well in static or low-motion regions; per-Gaussian volumetric deformation captures fast yet coherent flows; polynomial trajectory models favor globally smooth, low-curvature motion; and interpolation-based methods often yield locally diverse and irregular trajectories. Because real-world scenes typically contain a mixture of these motion patterns, no single deformation prior can perform optimally across all spatial and temporal regions. A more detailed, expert-specific motion analysis is provided in Appendix E.

Motivated by these insights, we propose Mixture of Experts for Dynamic Gaussian Splatting (MoE-GS), the first unified framework integrating multiple dynamic Gaussian models through a Mixture-of-Experts (MoE) architecture. Instead of relying on a single expert model, MoE-GS adaptively selects and blends multiple specialized experts, dynamically adjusting its decisions according to spatial, temporal, and scene-specific dynamics. It is important to clarify that, unlike sparsity-driven MoE architectures in large language models, MoE-GS is not intended to reduce FLOPs or memory usage. Our primary goal is to increase representational capacity and improve dynamic reconstruction quality by combining heterogeneous deformation priors; the efficiency techniques we introduce are complementary mechanisms to keep this additional cost practical.

A key innovation of MoE-GS is the *Volume-aware Pixel Router*, which adaptively blends expert outputs with spatial and temporal coherence. Rather than assigning routing weights solely at the pixel or Gaussian level—either ignoring volumetric structure or facing optimization difficulties—our router projects Gaussian-level decisions into pixel space using differentiable weight splatting. This provides adaptive, per-pixel weighting that effectively captures temporal and view-dependent variations, ensuring stable optimization. While MoE-GS significantly improves rendering quality, its multi-expert inference inherently increases computational overhead. To address this, we explore two independent strategies. First, we propose single-pass multi-expert rendering and gate-aware Gaussian pruning, which improve runtime efficiency by eliminating redundant rasterization and removing low-contributing primitives. This strategy is particularly effective when the number of experts is small (e.g., $N = 2$ or $N = 3$). Second, we introduce a knowledge distillation strategy that trains individual expert models using ground-truth supervision and pseudo-labels generated by the optimized MoE model, weighted at the pixel level by the router's predictions. This allows each expert to approximate the performance of the full MoE-GS without modifying its architecture. The benefit of this approach becomes more evident as the number of experts increases (e.g., $N = 4$ or higher), where directly running the full MoE incurs substantial computational cost.

In summary, our contributions include:

- MoE-GS, the first dynamic Gaussian splatting framework employing a Mixture-of-Experts architecture, enabling robust and adaptive reconstruction across diverse dynamic scenes.
- A novel Volume-aware Pixel Router that integrates expert outputs through differentiable weight splatting, achieving spatially and temporally coherent adaptive blending.
- Efficiency of MoE-GS is improved through single-pass multi-expert rendering and gate-aware Gaussian pruning, while a separate knowledge distillation strategy trains individual experts with pseudo-labels from the MoE model, enhancing quality without modifying the architecture.

## 2 RELATED WORKS

### 2.1 DYNAMIC NOVEL VIEW SYNTHESIS

Dynamic Novel View Synthesis aims to reconstruct novel views from sparse observations of temporally varying scenes, a challenging yet crucial task in computer vision (Gao et al., 2021; Park et al., 2021). Effectively modeling complex scene dynamics often requires advanced temporal modeling techniques, which substantially increase computational complexity and memory usage.

Implicit methods address scene dynamics by learning deformation fields that map points from a canonical space to observed frames. These methods typically rely on MLPs but differ in how they embed spatial and temporal information. Wu et al. (Wu et al., 2024) (4DGaussians) employ coordinate-conditioned embeddings (similar to HexPlane-style features) combined with lightweight MLPs for deformation estimation. Bae et al. (Bae et al., 2024) (E-D3DGS) instead adopt per-Gaussian embeddings with multiple volumetric fields and a coarse-to-fine strategy to improve temporal modeling. In contrast, explicit methods directly parameterize temporal variations of Gaussians without relying on canonical deformation fields. Polynomial-based models such as STG (Li et al., 2024) describe trajectories with parametric functions, while interpolation-based models such as Ex4DGS (Lee et al., 2024) reconstruct dynamics by interpolating between keyframes. These approaches simplify optimization by avoiding canonical deformation fields, but often struggle with complex motion.

While these methods achieve state-of-the-art results for specific types of deformations, they lack the flexibility to generalize across diverse dynamic scenarios. In contrast, our MoE-GS framework integrates multiple Gaussian-based models within a mixture-of-experts architecture, dynamically selecting and blending expert outputs to adaptively reconstruct a wide range of scene dynamics.

### 2.2 MIXTURE OF EXPERTS

MoE is an ensemble learning technique where multiple expert models specialize in distinct subtasks, with a gating network dynamically selecting the most relevant experts per input instance. MoE architectures have demonstrated scalability and efficiency by introducing sparsity (Shazeer et al., 2017), enabling conditional computation to scale model capacity without excessive computational

**Mixture of Experts Dynamic Gaussian Splatting**

**Candidate Experts**

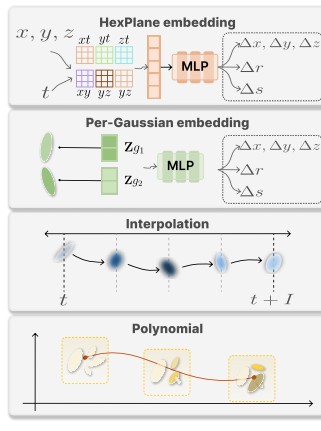

Figure 2: Overview of the MoE-GS framework. In Stage 1 (Expert Training), each expert is independently trained to reconstruct the dynamic scene by optimizing its own Gaussian representation, ensuring diverse modeling capabilities. In Stage 2 (Router Training), with all expert parameters fixed, the Volume-aware Pixel Router learns to dynamically blend expert-rendered images by computing spatially and temporally adaptive gating weights. The Candidate Experts (right) illustrate diverse Gaussian-based reconstruction methods integrated into our framework, including HexPlane Embedding-based, Per-Gaussian embedding-based, Interpolation-based, and Polynomial-based approaches, each suited for capturing different scene dynamics.

cost. This approach has been particularly successful in large-scale deep learning models, including large language models (LLMs) for machine translation (Lepikhin et al., 2021; Fedus et al., 2022) and computer vision applications such as multi-task learning (Ma et al., 2018), face forgery detection (Kong et al., 2022), and anomaly detection (Meng et al., 2024).

Inspired by MoE's ability to dynamically integrate specialized models, we propose applying MoE to Dynamic Gaussian Splatting, enabling a learnable expert selection process for dynamic scene reconstruction. Unlike existing methods, which statically select a single model for all frames, our MoE-GS learns to adaptively combine multiple experts based on scene characteristics. Furthermore, to mitigate the increased computational overhead associated with MoE, we introduce a knowledge distillation pipeline (Hinton et al., 2015; Xie et al., 2024), allowing experts to achieve near-MoE performance with significantly reduced computational cost.

## 3 METHOD

We propose a framework that closely follows the classic MoE structure, leveraging its proven effectiveness across diverse fields to address the problem of dynamic scene reconstruction.

### 3.1 PRELIMINARY

We briefly review the standard Mixture-of-Experts (MoE) architecture, which forms the basis for our proposed method. A standard MoE consists of multiple parallel expert networks $E_1, E_2, \ldots, E_N$ and a *Router* that adaptively combines expert outputs based on the input. Formally, given an input $x$, the MoE output is computed as follows:

$$\text{MoE}(x) = \sum_{k=1}^{N} G_k(x) \cdot E_k(x), \tag{1}$$

where $E_k(x)$ is the output of the $k$-th expert, and $G_k(x)$ represents the corresponding *gating weight* computed by the Router. The gating weights are typically computed by a Router, often implemented as a lightweight neural network (e.g., linear layer or MLP), defined as:

$$G_k(x) = \text{Softmax}(R_k(x)), \tag{2}$$

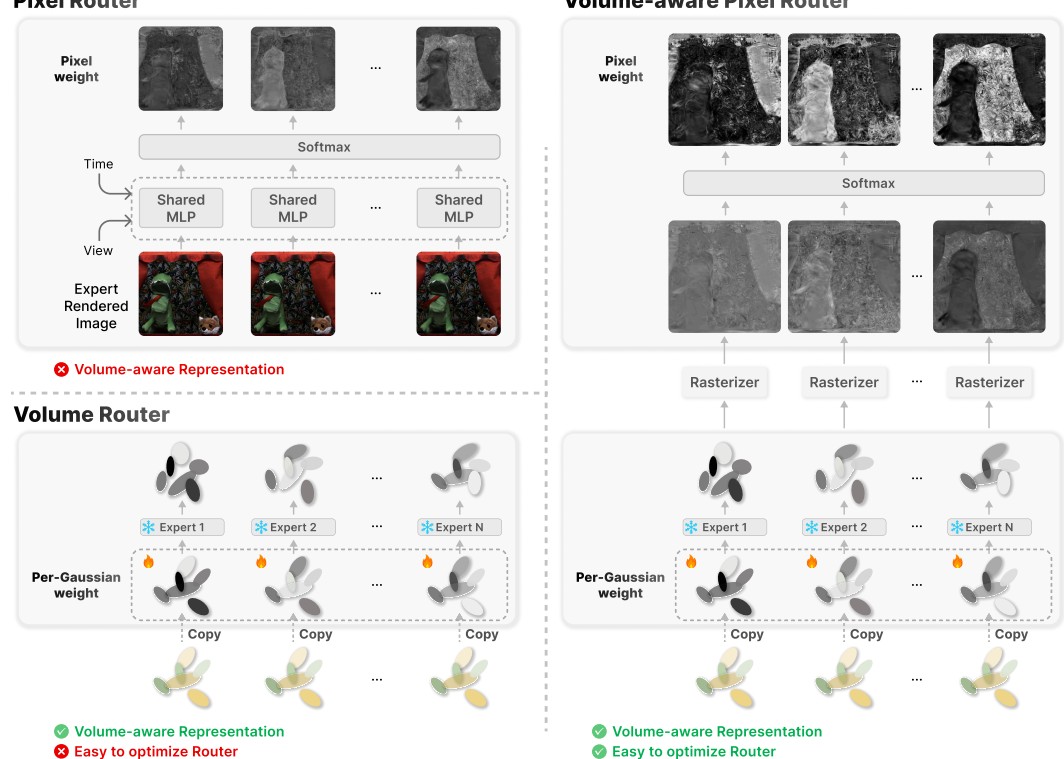

Figure 3: Comparison of Router Architectures. The Pixel Router (top-left) assigns weights purely at the pixel level, ignoring volumetric features. The Volume Router (bottom-left) uses Gaussian-level weights but is difficult to optimize. Our Volume-aware Pixel Router (right) combines Gaussian-level weights with rasterization-based splatting.

where $R_k(x)$ is the router output for the $k$-th expert, computed as:

$$R_k(x) = x^\top W_{r,k}, \tag{3}$$

with $W_{r,k}$ being the $k$-th column vector of trainable weight matrix $W_r$. This structure enables adaptive blending of experts based on input characteristics, enhancing performance and flexibility.

In the following sections, we detail how we extend this basic MoE architecture for dynamic scene reconstruction using Gaussian-based representations.

### 3.2 MIXTURE OF EXPERTS FOR DYNAMIC GAUSSIAN SPLATTING

As illustrated in Fig. 2, we propose an adaptation of the MoE architecture to dynamic Gaussian splatting. Our framework integrates diverse Gaussian-based dynamic reconstruction models, each treated as an expert, allowing us to leverage their strengths to improve rendering fidelity. Unlike traditional MoE models operating solely in feature space, our approach performs expert selection and blending directly on rendered 2D images, enhancing flexibility and reconstruction quality across complex temporal and spatial dynamics.

**Volume-aware Pixel Router.** A simple router might determine gating weights based purely on pixel-level features extracted by an MLP conditioned on time and view direction (Fig. 3, top-left). However, without considering intrinsic Gaussian properties (position, rotation, scale, and opacity), this approach lacks volumetric awareness and struggles to accurately capture temporal and view-dependent variations.

Another approach assigns gating weights directly per Gaussian in 3D space (Fig. 3, bottom-left). Here, each Gaussian's learned opacity is modulated by gating weights before rasterization into the 2D image space. However, as demonstrated in our experiments, directly optimizing gating weights in the fixed 3D Gaussian domain is challenging due to the indirect and complex relation between Gaussian parameters and their resulting pixel contributions.

To address these issues, we introduce the *Volume-aware Pixel Router* (Fig. 3, right), featuring three innovations: (1) explicit temporal and view-dependent encoding through learnable per-Gaussian weights; (2) intrinsic volumetric awareness via Gaussian attributes, enabling better expert differentiation; (3) rasterization-based weight splatting into pixel space, facilitating pixel-level adaptive blending informed by Gaussian-level features.

This approach effectively combines volumetric richness with temporal-view consistency, enabling stable optimization and high-quality rendering.

**Per-Gaussian Weight.** Gaussian Splatting (Kerbl et al., 2023) represents a scene as a set of Gaussians, each Gaussian $\mathcal{G}_i$ parameterized by position $\mu_i$, rotation $R_i$, and scale $S_i$:

$$\mathcal{G}_i = e^{-\frac{1}{2}(\mathbf{x}-\mu_i)^T \Sigma_i^{-1}(\mathbf{x}-\mu_i)}, \tag{4}$$

where $\Sigma_i = R_i S_i S_i^T R_i^T$. During splatting, overlapping Gaussians at each pixel are depth-ordered, and the pixel color $C$ is computed by combining each Gaussian's opacity $\sigma_i$ and color $\boldsymbol{c}_i$:

$$C = \sum_{i \in N} \boldsymbol{c}_i \alpha_i \prod_{j=1}^{i-1}(1 - \alpha_j), \quad \text{with} \quad \alpha_i = \sigma_i \mathcal{G}_i. \tag{5}$$

To construct our router, we duplicate each Gaussian $\mathcal{G}_i$ and replace its color attribute $\boldsymbol{c}_i$ with learnable per-Gaussian weights encoding temporal and view-dependent variations:

$$\boldsymbol{w}_i^{per} = [w_i, w_i^{dir}, (t \cdot w_i^{time})]^T, \tag{6}$$

where each scalar adapts dynamically according to the viewing direction and time step $t$.

**Adaptive Expert Gating via Weight Splatting.** Inspired by STG (Li et al., 2024), which uses feature splatting to compactly encode view-dependent radiance, we extend this concept by introducing rasterization-based weight splatting for adaptive expert gating. In contrast to STG's color refinement, our method dynamically regulates expert contributions through per-Gaussian weights. Each 3D per-Gaussian weight $w_i^{per}$ in Eq. (6) is projected onto the image plane through the differentiable Gaussian rasterizer, producing pixel-aligned embeddings $w_{2D}(u)$, $w_{2D}^{dir}(u)$, $w_{2D}^{time}(u)$ that aggregate the contributions of Gaussians overlapping pixel $u$. Specifically, these rasterized pixel-level weights are then refined by a lightweight MLP $\Phi$:

$$R'(u) = w_{2D}(u) + \Phi\big(w_{2D}^{dir}(u),\ w_{2D}^{time}(u),\ r(u)\big), \tag{7}$$

where $r(u)$ is the pixel viewing direction. Expert gating weights $G_k'$ are computed via softmax, then used to blend expert-rendered outputs $I_{E_k}$:

$$I_{MoE}(u) = \sum_{k=1}^{N} G_k'(u)\, I_{E_k}(u), \qquad G_k'(u) = \text{Softmax}(R_k'(u)). \tag{8}$$

We optimize Router parameters ($\boldsymbol{w}_i^{per}$ and $\Phi$) using standard Gaussian Splatting loss terms (L1 and SSIM). Given varying expert convergence rates, we adopt a two-stage training strategy to ensure stable and efficient optimization: experts are first optimized independently, and then the router is trained with these fixed experts (details in Appendix B.1).

**Gaussian-Level Interpretation of Pixel Gating.** Although the router outputs per-pixel weights, these gating decisions originate from per-Gaussian 3D signals (depth, visibility, deformation) and therefore carry 3D geometric structure. Because the weights are defined before rasterization, they can be naturally lifted back to the Gaussian domain in a responsibility-weighted manner, enabling simple post-hoc 3D fusion. The responsibility-based lifting procedure is provided in Appendix D.

### 3.3 Rendering and Pruning for Efficiency

MoE-GS improves fidelity by leveraging multiple experts but introduces inefficiency—namely, repeated rasterization and many low-contribution Gaussians. To address this, we propose two techniques: **Single-Pass Multi-Expert Rendering**, which shares projection and rasterization across all experts, and **Gate-Aware Gaussian Pruning**, which removes Gaussians with negligible influence.

**Single-Pass Multi-Expert Rendering.** In the baseline pipeline each expert is rasterized independently, leading to repeated projection and visibility computation across $K$ passes. We address this redundancy by merging all Gaussians into a single batch and assign each Gaussian a one-hot expert identity $e_j \in \mathbb{R}^K$. The expert-specific color at pixel $u$ is computed as

$$C_k(u) = \sum_{j=1}^{M} T_j(u)\, \alpha_j(u)\, c_j \cdot (e_j)_k, \tag{9}$$

where $M$ is the total number of Gaussians across experts, $T_j(u) = \prod_{m=1}^{j-1}(1 - \alpha_m(u))$ is the transmittance, $\alpha_j$ the opacity, and $(e_j)_k$ selects Gaussians of expert $k$. This design computes projection and visibility only once for all Gaussians, while expert-specific outputs are separated during alpha blending. As a result, redundant kernel launches and memory traversals inherent in the multi-pass pipeline are eliminated, improving GPU utilization without altering the rendering formulation.

**Gate-Aware Gaussian Pruning.** Independently trained experts often produce overlapping Gaussians with low contribution. To selectively remove these, we accumulate the gradient of gating weights $G'_k$ with respect to the per-Gaussian weights $\boldsymbol{w}_i^{per}$, measuring how strongly each Gaussian influences the MoE image. The importance of Gaussian $i$ across all training views $\mathcal{D}$ is computed as

$$\mathcal{E}_i = \frac{1}{|\mathcal{D}|} \sum_{v \in \mathcal{D}} \left\| \frac{\partial G'_k(v)}{\partial \boldsymbol{w}_i^{per}(v)} \right\|. \tag{10}$$

Gaussians with $\mathcal{E}_i < \tau$ are progressively pruned, yielding a compact yet faithful representation. This strategy is effective because Gaussians with negligible gradients consistently show little impact on the gating weights and thus contribute minimally to the final image. By removing only these Gaussians, the model reduces rendering cost without sacrificing visual fidelity, unlike naive ratio-based pruning which destabilizes optimization.

However, aggressive pruning alone is insufficient: as the number of experts increases, the total Gaussian count grows proportionally, and naively discarding large portions destabilizes optimization. To overcome this limitation, we introduce a complementary distillation strategy.

### 3.4 DISTILLATION-BASED EXPERT TRAINING

To further reduce inference cost while maintaining stability, we adopt knowledge distillation (Hinton et al., 2015; Xie et al., 2024). Each expert $E_k$ is trained from scratch using both ground-truth images and MoE outputs as supervision. We use the MoE-rendered image $I_{MoE}$ as pseudo-ground truth and routing weights $G'_k$ as confidence scores. The distillation loss is

$$\mathcal{L}_k^{\text{KD}} = \lambda \cdot \mathcal{L}(G'_k \cdot I_{E_k},\, G'_k \cdot I_{GT}) + (1 - \lambda) \cdot \mathcal{L}((1 - G'_k) \cdot I_{E_k},\, (1 - G'_k) \cdot I_{MoE}), \tag{11}$$

where $\mathcal{L}$ combines L1 and SSIM losses, and $\lambda$ balances ground-truth vs. MoE supervision. This encourages each expert to specialize in reliable regions guided by ground truth while leveraging MoE outputs in uncertain areas. As a result, individual experts approximate the performance of the full MoE-GS model with significantly reduced complexity, enabling efficient real-time deployment.

## 4 EXPERIMENTS

This section evaluates the effectiveness of MoE-GS through experiments assessing its generalization, scalability, and deployability. We present results on MoE-GS, including comparisons with baselines and detailed analyses of expert configurations. We further conduct ablation studies to first examine the contribution of the MoE architecture, specifically the individual experts and the routing mechanism. Next, we analyze the impact of our proposed efficiency strategies, such as single-pass rendering and pruning. Finally, we investigate the role and impact of our distillation strategy on the model's overall performance.

### 4.1 EXPERIMENTAL SETUP

**Implementation Details.** To evaluate the robustness of MoE-GS under diverse real-world deformations, we conduct experiments on two standard benchmarks for dynamic scene reconstruction: Neural 3D Video (N3V) (Li et al., 2022) and Technicolor (Sabater et al., 2017). Per-Gaussian weights are

Table 1: Performance comparison on the N3V dataset (Li et al., 2022). †: Models were trained on a dataset split into 150 frames. We highlight best and second-best values for each metric.

| Model | PSNR (dB) ↑ | | | | | | |
|---|---|---|---|---|---|---|---|
| | Coffee Martini | Cook Spinach | Cut Roasted Beef | Flame Salmon | Flame Steak | Sear Steak | Average |
| HyperReel (Attal et al., 2023) | 28.37 | 32.30 | 32.92 | 28.26 | 32.20 | 32.57 | 31.10 |
| K-Planes (Fridovich-Keil et al., 2023) | 29.99 | 32.60 | 31.82 | 30.44 | 32.38 | 32.52 | 31.63 |
| MixVoxels-L (Wang et al., 2023) | 29.63 | 32.25 | 32.40 | 29.81 | 31.83 | 32.10 | 31.34 |
| 3DGStream (Sun et al., 2024) | 27.75 | 33.31 | 33.21 | 28.42 | 34.30 | 33.01 | 31.67 |
| DASS (Liu et al., 2024) | 28.15 | 33.83 | 33.54 | 28.84 | 34.26 | 33.33 | 31.99 |
| SaRO-GS (Yan et al., 2024) | 28.96 | 33.19 | 33.91 | 29.14 | 33.83 | 33.89 | 32.15 |
| SwinGS (Liu & Banerjee, 2024) | 27.99 | 33.66 | 34.03 | 28.24 | 32.94 | 33.32 | 31.69 |
| 4DGaussians (Wu et al., 2024) | 29.09 | 32.78 | 33.15 | 29.76 | 31.81 | 32.01 | 31.43 |
| E-D3DGS (Bae et al., 2024) | 30.04 | 33.11 | 33.85 | 30.49 | 32.77 | 33.70 | 32.33 |
| STG† (Li et al., 2024) | 28.16 | 33.09 | 34.15 | 29.09 | 33.25 | 33.77 | 31.92 |
| Ex4DGS (Lee et al., 2024) | 28.72 | 33.24 | 33.73 | 29.33 | 33.91 | 33.69 | 32.10 |
| **MoE-GS (N=2)** | 30.27 | 33.43 | 34.05 | 30.66 | 32.92 | 33.90 | 32.54 |
| **MoE-GS (N=3)** | 30.27 | 33.86 | 34.90 | 30.92 | 34.52 | 34.88 | 33.23 |
| **MoE-GS (N=4)** | 30.43 | 34.24 | 35.20 | 30.92 | 34.38 | 34.42 | 33.27 |

Table 2: Comparison results on the Technicolor dataset (Sabater et al., 2017).

| Model | PSNR (dB) ↑ | | | | | |
|---|---|---|---|---|---|---|
| | Birthday | Fabien | Painter | Theater | Train | Average |
| DyNeRF (Li et al., 2022) | 29.20 | 32.76 | 35.95 | 29.53 | 31.58 | 31.80 |
| HyperReel (Attal et al., 2023) | 29.99 | 34.70 | 35.91 | 33.32 | 29.74 | 32.73 |
| 4DGaussians (Wu et al., 2024) | 30.87 | 33.56 | 34.36 | 29.81 | 25.35 | 30.79 |
| STG (Li et al., 2024) | 32.16 | 35.70 | 37.18 | 31.00 | 32.39 | 33.69 |
| E-D3DGS (Bae et al., 2024) | 32.38 | 34.24 | 36.20 | 31.10 | 31.37 | 33.06 |
| Ex4DGS (Lee et al., 2024) | 32.35 | 35.18 | 36.60 | 31.77 | 31.37 | 33.45 |
| **MoE-GS (N=3)** | 33.26 | 36.26 | 37.63 | 32.88 | 32.89 | 34.55 |

optimized with the RAdam optimizer (Liu et al., 2019), using a learning rate of 0.5. Experiments are performed on NVIDIA A6000 GPUs. Additional implementation details are provided in Appendix 5.

**MoE-GS Expert Configurations.** For clarity and reproducibility, we fix the expert sets used in all experiments. We evaluate MoE-GS with $N = 2, 3, 4$ experts using the following heterogeneous combinations: $N = 2$: {Ex4DGS (Lee et al., 2024), STG (Li et al., 2024)}; $N = 3$: + E-D3DGS (Bae et al., 2024); $N = 4$: + 4DGaussians (Wu et al., 2024). These fixed sets cover diverse deformation families, and are used consistently across all experiments. We note that this design choice is for clarity; MoE-GS itself remains deformation-agnostic.

## 4.2 RESULTS ON MoE-GS

**Baselines.** The expert set in MoE-GS consists of multiple Gaussian-based dynamic reconstruction models, each capturing temporal deformations in a distinct way. We include embedding-based methods (4DGaussians (Wu et al., 2024), E-D3DGS (Bae et al., 2024)), an interpolation-based model (Ex4DGS (Lee et al., 2024)), and a polynomial-based method (STG (Li et al., 2024)), ensuring diversity across deformation representations. For broader comparison, we also evaluate against representative NeRF-based baselines.

**Quantitative Evaluation.** Tables 1 and 2 present the per scene results on the N3V (Li et al., 2022) and Technicolor (Sabater et al., 2017) datasets. MoE-GS achieves state-of-the-art average performance on both datasets, consistently outperforming most baseline methods across individual scenes. This demonstrates its strong ability to generalize across diverse scenarios by selectively leveraging different experts. In addition, Table 3 presents an efficiency analysis of MoE-GS, showing that its inference and memory overhead can be significantly reduced via Gate-Aware Pruning. MoE-GS (N=2), which integrates STG (Li et al., 2024) and Ex4DGS (Lee et al., 2024) as experts, outperforms both in PSNR while exhibiting moderate computational overhead. This suggests that combining distinct expert models can enhance reconstruction quality without incurring excessive cost. Detailed per-metric results, as well as additional evaluations on the monocular HyperNeRF dataset (Park et al., 2021), are provided in the Appendix C.1.

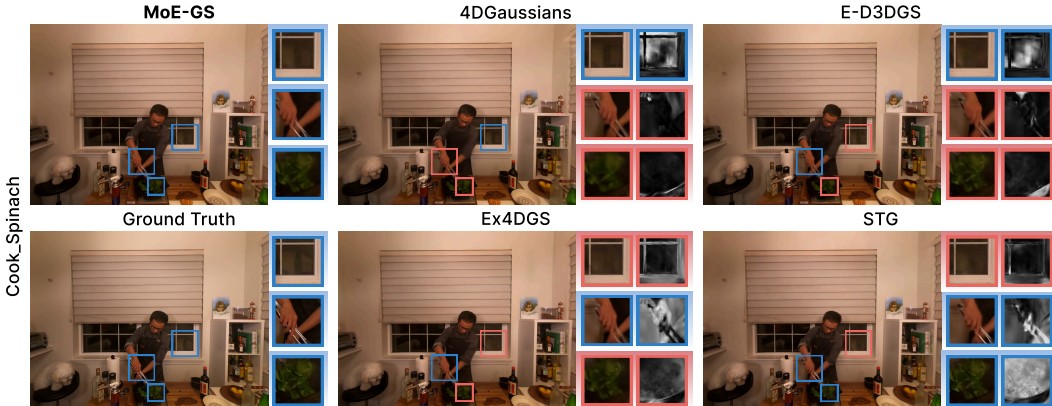

Figure 4: **N3V Qualitative Results** Comparison of our MoE-GS with other dynamic Gaussian splatting methods on Neural 3D Video dataset (Li et al., 2022). black background highlight the method that produces the most visually accurate result among the baselines for each region.

**Qualitative Evaluation.** As shown in Figure 4, we present the gating weights predicted by the router, the outputs of individual experts, and the resulting MoE-GS image synthesized through their combination. These expert contributions align closely with the router's predictions, indicating that MoE-GS effectively blends experts in a spatially adaptive manner. Additional qualitative results are provided in the Appendix C.2

Table 3: Efficiency evaluation with N=2 Expert Variants on N3V (Li et al., 2022).

| Model | PSNR ↑ | FPS ↑ | Memory ↓ |
|---|---|---|---|
| STG† (Li et al., 2024) | 31.92 | 88.5 | 609.5 |
| Ex4DGS (Lee et al., 2024) | 32.01 | 120 | 122.8 |
| MoE-GS (N=2) | 32.82 | 44 | 878.7 |
| MoE-GS ( 55% pruning) | 32.80 | 83 | 351.2 |
| MoE-GS ( 75% pruning) | 32.45 | 101 | 281.3 |

**Analysis of MoE Configurations.** We conduct ablation studies to evaluate the design choices in MoE-GS, focusing on the architecture of the MoE router and efficiency optimizations.

**1) MoE Router Variants** We compared three router architectures for integrating expert outputs: Pixel Router, Volume Router, and our proposed Volume-aware Pixel Router. *Pixel Router* performs blending in 2D image space using a lightweight MLP, which supports stable optimization in image space. However, as shown in Table 4 and Figure 5, it underperforms in quan-

Table 4: Performance Comparison of Different MoE Router Variants

| Model | PSNR ↑ | SSIM ↑ | LPIPS ↓ |
|---|---|---|---|
| Pixel Router | 31.12 | 0.952 | 0.022 |
| Volume Router | 32.05 | 0.951 | 0.022 |
| Volume-aware Pixel Router | **33.23** | **0.954** | **0.021** |

titative metrics and produces overly smooth results, failing to capture sharp boundaries, likely due to its lack of 3D spatial context. *Volume Router*, which blends expert Gaussians in 3D space by adjusting their opacities, better preserves geometric structure but underperforms in detail fidelity. We observed frequent optimization instability and oversmoothing artifacts, especially in regions with fine textures. In contrast, our *Volume-aware Pixel Router* achieves a better balance: it maintains sharp structural details and consistently outperforms other variants both quantitatively and qualitatively. We attribute this to its 2D blending formulation, which supports stable training, combined with its use of 3D-aware weight splatting during training that injects geometric context into the routing process.

**2) Efficiency Optimizations** We evaluate the effectiveness of our efficiency techniques—Single-Pass Multi-Expert Rendering and Gate-Aware Gaussian Pruning—by measuring their impact on PSNR, FPS, and memory usage. Without Single-Pass, each expert is rasterized independently, repeating projection and blending steps

Table 5: Ablation of Efficiency Optimizations.

| Model | PSNR ↑ | FPS ↑ | Memory ↓ |
|---|---|---|---|
| w/o Single-Pass & Pruning | 32.54 | 36 | 747 |
| w/o Single-Pass | 33.23 | 40 | **270** |
| w/o Pruning | 32.54 | 60 | 747 |
| **MoE-GS (N=3)** | **33.23** | 68 | **270** |

and leading to significant FPS drops due to GPU overhead (Table 3). Without pruning, all Gaussians are retained—including those with negligible contributions—which preserves reconstruction quality but reduces rendering and memory efficiency. When both techniques are removed, the baseline suffers from compounded overhead in rendering and memory usage. In contrast, our full model (MoE-GS, N=3) incorporates both strategies, eliminating redundant computation and discarding uninformative

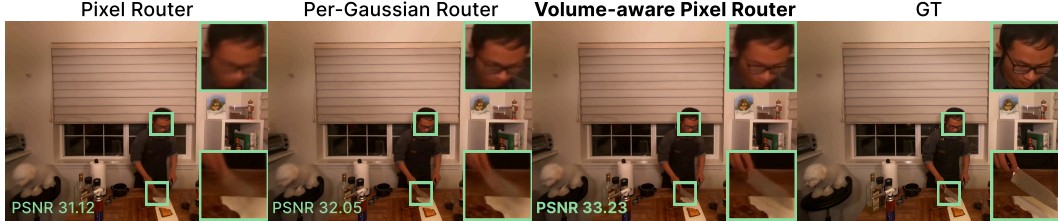

Figure 5: **Qualitative comparison of MoE Router variants.** Each router applies a different routing strategy, resulting in varied structural consistency and detail across the outputs.

Gaussians. As a result, it achieves comparable PSNR to the baseline while significantly improving FPS and reducing memory footprint (Table 5).

**Analysis of training Cost.** Since MoE-GS employs multiple dynamic GS experts, we analyze how expert training budget influences overall performance. In principle, using $N$ experts could increase training time by a factor of $N$, but we find that MoE-GS remains effective even when experts are trained with significantly reduced budgets.

**1) Partial expert training.** To quantify the practical training overhead of MoE-GS, we experiment with reducing the training budget of each expert. Here, a budget of 100% corresponds to the expert's full training time (approximately 4.2 hours for E-D3DGS, 2.2 hours for STG, and 1.5 hours for 4DGaussians), while 50%, 20%, and 10% correspond

Table 6: Effect of expert training budget on MoE-GS (N=3).

| Model | 100% | 50% | 20% | 10% |
|---|---|---|---|---|
| E-D3DGS (Bae et al., 2024) | 32.33 | 32.19 | 31.87 | 30.60 |
| STG (Li et al., 2024) | 31.92 | 31.76 | 31.41 | 31.19 |
| 4DGaussians (Wu et al., 2024) | 31.43 | 31.02 | 30.90 | 30.64 |
| **MoE-GS (N=3)** | **33.23** | **32.71** | **32.60** | **32.14** |

to proportional reductions in wall-clock time. Thus, the budget axis directly reflects the actual training time per expert. Table 6 shows that even when each expert is trained with only 20% of its usual budget (e.g., reducing a 4.2-hour expert to about 50 minutes, or a 1.5-hour expert to under 20 minutes), MoE-GS still outperforms fully trained single-expert baselines. This suggests that MoE-GS does not rely on fully converged experts and that meaningful gains can be achieved with substantially reduced per-expert training time.

**2) Router overhead.** The router itself is lightweight and adds less than 5% computation relative to expert training, making its additional cost negligible. These results show that MoE-GS does not require fully converged experts to achieve performance, and that its practical training overhead is far lower than linearly scaling with the number of experts.

### 4.3    DISTILLATION-BASED EXPERT TRAINING

We compared expert models retrained from scratch with distilled versions trained under identical settings, differing only in supervision via MoE-generated images and routing weights. Table 7 shows a consistent improvement in PSNR, SSIM, LPIPS across all expert types, indicating the effectiveness of MoE-guided supervision. This demonstrates that when the number of experts grows large, applying distillation to each expert remains an effective way to achieve sig-

Table 7: Ablation Studies on MoE-GS Distillation Methods on Technicolor dataset

| Model | PSNR ↑ | SSIM ↑ | LPIPS ↓ |
|---|---|---|---|
| E-D3DGS (Bae et al., 2024) | 32.88 | 0.902 | 0.111 |
| E-D3DGS (Bae et al., 2024) (Distilled) | 33.67 | 0.915 | 0.091 |
| STG (Li et al., 2024) | 32.83 | 0.915 | 0.083 |
| STG (Li et al., 2024) (Distilled) | 33.10 | 0.917 | 0.082 |
| Ex4DGS (Lee et al., 2024) | 33.57 | 0.918 | 0.086 |
| Ex4DGS (Lee et al., 2024) (Distilled) | 33.91 | 0.923 | 0.079 |

nificant performance gains. Qualitative results and weighting ablations are provided in Appendix C.2 and G.

### 5    CONCLUSION

We propose MoE-GS, the first Mixture of Experts framework for dynamic Gaussian splatting, enabling adaptive and high-fidelity scene reconstruction. Our method introduces a Volume-aware Pixel Router that combines pixel-based and volumetric expert blending while reducing computational cost through knowledge distillation. The per-Gaussian responsibilities exposed by MoE-GS also provide a useful signal for future canonical-space fusion or refinement methods seeking more explicit geometric unification. Overall, MoE-GS offers a scalable and effective framework for dynamic Gaussian Splatting, improving 4D consistency while remaining compatible with evolving GS representations.

ACKNOWLEDGMENT

This research was supported by the National Research Foundation of Korea (NRF) grant funded by the Korea government (MSIT) (RS-2024-00414230, RS-2024-00456152) and the Ministry of Education (MOE) (RS-2025-25423291). This work was also supported by the Regional Innovation System & Education (RISE) program through the institute for Regional Innovation System & Education in Busan Metropolitan City, funded by the Ministry of Education (MOE) and the Busan Metropolitan City, Republic of Korea (2025-RISE-02-004-12880001-05). The computational resources were provided by the Cluster Server for Computational Science at Pusan National University and the National IT Industry Promotion Agency (NIPA) (G2025-0403).

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

APPENDIX

## A  OVERVIEW

This appendix includes detailed implementation information in Appendix B, additional quantitative and qualitative results in Appendix C, Gaussian-Level Interpretation and Geometry Evaluation Appendix D, analysis of expert specialization in Appendix E, Best-of-N Retraining Appendix F, Ablation on Distillation Weighting Strategies Appendix G.

### A.1  REFERENCED IN THE MAIN PAPER

The sections listed below are directly referenced in the main paper for further details:

- Optimization strategy of MoE-GS (Appendix B.1)
- Implementation details of Expert Training (Appendix B.2)
- Implementation details of Router Training (Appendix B.3)
- Implementation details of Single-Pass Multi-Expert Rendering (Appendix B.4)
- Quantiative Results (Appendix C.1)
- Qualitative Results (Appendix C.2)
- Per-Gaussian Contributions and Responsibilities (Appendix D.1)
- Post-hoc Gaussian Fusion using Lifting Weights (Appendix D.2)
- Multi-view Depth Consistency Evaluation (Appendix D.3)
- Analysis of Expert Specialization (Appendix E)
- Best-of-N Retraining (Appendix F)
- Ablation on Distillation Weighting Strategies (Appendix G)

## B  IMPLEMENTATION DETAILS

This section provides implementation and training details of the proposed MoE-GS framework. We first describe the overall two-stage training strategy, which separates expert model training from router optimization to ensure stable convergence and prevent dominance by faster-converging experts. Next, we present the training setup for individual expert models, including initialization and baseline-aligned hyperparameters. Finally, we outline architectural details of the proposed Volume-aware Pixel Router, which enables spatially and temporally coherent expert blending in the MoE-GS pipeline.

### B.1  TWO-STAGE TRAINING STRATEGY

To ensure stable convergence and balanced optimization, we adopt a two-stage training strategy that decouples expert training from router optimization. Jointly training both components can lead to suboptimal convergence, as faster-converging experts tend to dominate the gating process early on, leaving others underutilized and under-optimized. To mitigate this, we first train each expert model independently to ensure that it can reconstruct the entire scene without relying on other experts. Once trained, all expert parameters are frozen. In the second stage, we optimize the routing components—specifically the per-Gaussian parameters $w_i, w_i^{dir}, w_i^{time}$ and the MLP $\Phi$—to learn an adaptive gating strategy that dynamically selects and blends experts based on spatial, temporal, and view-dependent cues.

### B.2  STAGE 1: EXPERT TRAINING

In the first step of MoE-GS training, each expert model is independently optimized before integration into the MoE framework. Since MoE-GS reconstructs dynamic scenes by blending the outputs of multiple experts, it is critical that each expert achieves its best possible performance. To this end, we retain the original training strategies proposed in their respective works without modification. All expert models are initialized using point clouds generated by COLMAP.

- Ex4DGS (Lee et al., 2024) is initialized with sparse point clouds obtained via Structure-from-Motion (SfM).

- 4DGaussians (Wu et al., 2024) and E-D3DGS(Bae et al., 2024) are initialized with down-sampled versions of these dense point clouds.

- For STG (Li et al., 2024), we follow its original strategy, merging point clouds from all frames to obtain a globally consistent initialization that serves as a strong prior for optimizing Gaussian attributes.

Each expert is trained using its original learning rate schedule, as the hyperparameters are specifically tuned to each model's architecture and deformation representation.

## B.3 STAGE 2: ROUTER TRAINING

In the second stage of training, we optimize the Volume-aware Pixel Router while keeping all expert models frozen. This allows the router to focus solely on learning effective expert blending strategies without being influenced by the convergence rate of individual experts. Specifically, we optimize per-Gaussian parameters $w_i, w_i^{dir}, w_i^{time}$ and the MLP $\Phi$, which collectively determine the final expert weights for blending. All three parameters are fully learnable. To avoid introducing any handcrafted directional or temporal bias, we initialize $w_i^{dir}$ and $w_i^{time}$ using neutral near-zero constants, allowing the router to gradually learn meaningful directional and temporal sensitivities directly from data. Joint training of experts and the router often leads to suboptimal expert utilization, as rapidly converging experts can dominate the early stages of routing. Our two-stage approach avoids this issue by ensuring that all experts are first trained to full capacity, allowing the router to later learn how to combine their outputs most effectively.

**Volume-aware Pixel Router.** To generate routing weights, our router begins with the per-Gaussian weights $w_i^{per}$ defined in 3D space. As illustrated in Figure 6, each expert's Gaussian is duplicated, and its color attribute is replaced with a learnable weights $w_i^{per}$. The remaining attributes—such as position, scale, and rotation—are directly copied from the pre-trained expert Gaussians. This enables $w_i^{per}$ to make the router aware of the expert's volumetric structure, allowing the 2D splatted weights to reflect how each expert behaves under different viewing directions and time steps. To further model view- and time-dependent variation, we also splat auxiliary features $w_{2D}^{dir}, w_{2D}^{time}$, and the ray direction $r$, which are fed into a lightweight MLP. The output of this MLP is added to $w_{2D}$ in a residual manner to produce the final routing weights.

Figure 6: Architectural details of the Volume-aware Pixel Router.

To promote spatial coherence, we adopt a convolutional MLP architecture that leverages local pixel context. For computational efficiency, this MLP is shared across all experts. We empirically set the learning rate to 0.05 for the shared MLP and $w_i^{time}$, and to 0.5 for $w_i$ and $w_i^{dir}$. This modular and coherent design enables the router to learn adaptive, per-pixel expert blending strategies that generalize effectively across diverse dynamic scenes.

## B.4 SINGLE-PASS MULTI-EXPERT RENDERING

To efficiently deploy MoE-GS, we complement our design with an optimized rendering pipeline. In particular, independently rasterizing each expert triggers repeated kernel launches and separate memory traversal over each expert's Gaussian buffer, significantly increasing memory IO pressure. To address this bottleneck, we adopt a Single-Pass Multi-Expert Rendering strategy, which processes all Gaussians in a single batched pass and eliminates these redundant kernel launches while still preserving expert-specific outputs. For completeness, we also investigated two alternative implementations for rendering multiple experts: (1) sequential expert execution with CPU offloading, and (2) distributing experts across multiple GPUs.

**Sequential execution** becomes prohibitively slow due to repeated PCIe transfers between the host and device. Gaussian Splatting requires high-frequency random access to Gaussian attributes (positions, scales, SH coefficients, opacities) during splatting. Moving these buffers back and forth over PCIe significantly increases latency and prevents any effective kernel fusion, leading to very large slowdowns in practice.

**Multi-GPU distribution** also provides limited benefit. Each expert resides in a heterogeneous and non-alignable 3D deformation space, so distributing experts across GPUs requires duplicating each expert's Gaussian buffer as well as synchronizing per-pixel accumulations across devices. This synchronization step introduces substantial inter-GPU communication overhead, often outweighing the parallelism benefits and resulting in negligible wall-clock speedup.

These observations motivate our choice of the Single-Pass Multi-Expert Rendering strategy as the most stable and efficient trade-off for MoE-GS workloads.

## C ADDITIONAL QUANTITATIVE AND QUALITATIVE RESULTS

### C.1 QUANTITATIVE RESULTS

To comprehensively evaluate MoE-GS across perceptual, structural, and pixel-wise metrics, Tables 8 and 9 present per-scene quantitative results on the N3V (Li et al., 2022) and Technicolor (Sabater et al., 2017) datasets. Table 8 reports N3V results using a 4-expert configuration, where MoE-GS consistently outperforms individual experts across all scenes and metrics, demonstrating strong generalization. Table 9 shows results on Technicolor with a 3-expert setup using E-D3DGS (Bae et al., 2024), STG (Li et al., 2024), and Ex4DGS (Lee et al., 2024). MoE-GS maintains strong performance even with fewer experts, achieving top or near-top performance across most scenes and the highest overall average across all evaluation metrics.

To assess its scalability to monocular settings, Table 10 evaluates MoE-GS on the HyperNeRF dataset (Park et al., 2021). Despite the limited input views, MoE-GS transfers well to this setting, maintaining high fidelity and demonstrating strong adaptability beyond the multi-view reconstruction scenario.

In addition, Table 11 presents an ablation study of our distillation strategies on the Technicolor dataset, reporting per-scene results and highlighting the contribution of each component to final performance.

Table 8: Additional Quantitative Results on Experts and MoE-GS for the N3V Dataset (Li et al., 2022). †: Models were trained on a dataset split into 150 frames. We highlight best and second-best values for each metric.

| Model | Coffee Martini | | | Cook Spinach | | | Cut Roast Beef | | |
|---|---|---|---|---|---|---|---|---|---|
| | PSNR | SSIM | LPIPS | PSNR | SSIM | LPIPS | PSNR | SSIM | LPIPS |
| 4DGaussians (Wu et al., 2024) | 29.09 | 0.923 | 0.066 | 32.78 | 0.955 | 0.041 | 33.15 | 0.954 | 0.048 |
| E-D3DGS (Bae et al., 2024) | 30.04 | 0.930 | 0.058 | 33.11 | 0.961 | 0.041 | 33.85 | 0.958 | 0.042 |
| STG † (Li et al., 2024) | 28.16 | 0.927 | 0.061 | 33.09 | 0.961 | 0.033 | 34.15 | 0.964 | 0.032 |
| Ex4DGS (Lee et al., 2024) | 28.72 | 0.918 | 0.070 | 33.24 | 0.956 | 0.042 | 33.73 | 0.958 | 0.040 |
| MoE-GS (N=4) | 30.43 | 0.940 | 0.054 | 34.24 | 0.966 | 0.031 | 35.08 | 0.968 | 0.030 |

| Model | Flame Salmon | | | Flame Steak | | | Sear Steak | | |
|---|---|---|---|---|---|---|---|---|---|
| | PSNR | SSIM | LPIPS | PSNR | SSIM | LPIPS | PSNR | SSIM | LPIPS |
| 4DGaussians (Wu et al., 2024) | 29.76 | 0.928 | 0.062 | 31.81 | 0.962 | 0.032 | 32.01 | 0.964 | 0.032 |
| E-D3DGS (Bae et al., 2024) | 30.49 | 0.936 | 0.054 | 32.77 | 0.960 | 0.037 | 33.70 | 0.964 | 0.033 |
| STG † (Li et al., 2024) | 29.09 | 0.928 | 0.057 | 33.25 | 0.968 | 0.026 | 33.77 | 0.969 | 0.026 |
| Ex4DGS (Lee et al., 2024) | 29.33 | 0.925 | 0.066 | 33.91 | 0.963 | 0.034 | 33.69 | 0.960 | 0.035 |
| MoE-GS (N=4) | 30.92 | 0.942 | 0.049 | 34.38 | 0.972 | 0.026 | 34.42 | 0.972 | 0.026 |

Table 9: Additional Quantitative Results on Experts and MoE-GS for the Technicolor Dataset (Sabater et al., 2017).

| Model | Birthday | | | Fabien | | | Painter | | |
|---|---|---|---|---|---|---|---|---|---|
| | PSNR | SSIM | LPIPS | PSNR | SSIM | LPIPS | PSNR | SSIM | LPIPS |
| DyNeRF (Li et al., 2022) | 29.20 | N/A | 0.067 | 32.76 | N/A | 0.242 | 35.95 | N/A | 0.146 |
| HyperReel (Attal et al., 2023) | 29.99 | 0.922 | 0.053 | 34.70 | 0.895 | 0.186 | 35.91 | 0.923 | 0.117 |
| 4DGaussians (Wu et al., 2024) | 30.87 | 0.904 | 0.087 | 33.56 | 0.854 | 0.186 | 34.36 | 0.884 | 0.136 |
| STG (Li et al., 2024) | 31.90 | 0.940 | 0.044 | 35.70 | 0.904 | 0.114 | 37.07 | 0.928 | 0.093 |
| Ex4DGS (Lee et al., 2024) | 32.36 | 0.941 | 0.045 | 35.19 | 0.896 | 0.124 | 36.66 | 0.932 | 0.091 |
| MoE-GS (N=3) | 33.26 | 0.947 | 0.049 | 36.26 | 0.908 | 0.121 | 37.63 | 0.939 | 0.083 |

| Model | Theater | | | Train | | | Average | | |
|---|---|---|---|---|---|---|---|---|---|
| | PSNR | SSIM | LPIPS | PSNR | SSIM | LPIPS | PSNR | SSIM | LPIPS |
| DyNeRF (Li et al., 2022) | 29.53 | N/A | 0.188 | 31.58 | N/A | 0.067 | 31.80 | N/A | 0.142 |
| HyperReel (Attal et al., 2023) | 33.32 | 0.895 | 0.115 | 29.74 | 0.895 | 0.072 | 32.73 | 0.906 | 0.109 |
| 4DGaussians (Wu et al., 2024) | 29.81 | 0.841 | 0.155 | 25.35 | 0.730 | 0.166 | 30.79 | 0.843 | 0.146 |
| STG (Li et al., 2024) | 31.08 | 0.879 | 0.140 | 32.32 | 0.937 | 0.045 | 33.61 | 0.918 | 0.087 |
| Ex4DGS (Lee et al., 2024) | 31.79 | 0.882 | 0.130 | 31.39 | 0.928 | 0.055 | 33.48 | 0.916 | 0.089 |
| MoE-GS (N=3) | 32.88 | 0.900 | 0.115 | 32.89 | 0.944 | 0.046 | 34.58 | 0.928 | 0.083 |

Table 10: Comparison results on the HyperNeRF dataset (Park et al., 2021).

| Model | PSNR (dB) ↑ | | | | |
|---|---|---|---|---|---|
| | 3dprinter | banana | broom | chicken | Average |
| 4DGaussians (Wu et al., 2024) | 22.16 | 22.90 | 20.88 | 30.12 | 24.02 |
| E-D3DGS (Bae et al., 2024) | 22.41 | 23.38 | 20.07 | 29.11 | 23.74 |
| MoE-GS (N=2) | 22.84 | 24.75 | 21.26 | 30.37 | 24.81 |

Table 11: Ablation study on distillation strategies evaluating the effect of routing-weight-based adaptive supervision on the Technicolor dataset (Sabater et al., 2017).

| Model | Training | Birthday | | | Fabien | | | Painter | | |
|---|---|---|---|---|---|---|---|---|---|---|
| | | PSNR | SSIM | LPIPS | PSNR | SSIM | LPIPS | PSNR | SSIM | LPIPS |
| E-D3DGS (Bae et al., 2024) | Retrained (GT Loss) | 32.05 | 0.936 | 0.050 | 34.7 | 0.878 | 0.171 | 36.26 | 0.931 | 0.089 |
| | Distilled (w/o Weight) | 32.17 | 0.942 | 0.048 | 34.8 | 0.883 | 0.164 | 36.77 | 0.934 | 0.089 |
| | Distilled (Ours) | 32.24 | 0.946 | 0.038 | 35.68 | 0.902 | 0.120 | 37.20 | 0.939 | 0.078 |
| STG (Li et al., 2024) | Retrained (GT Loss) | 33.15 | 0.947 | 0.038 | 34.87 | 0.901 | 0.117 | 33.61 | 0.907 | 0.098 |
| | Distilled (w/o Weight) | 33.27 | 0.948 | 0.040 | 34.99 | 0.901 | 0.188 | 33.66 | 0.907 | 0.099 |
| | Distilled (Ours) | 33.46 | 0.949 | 0.039 | 35.01 | 0.902 | 0.117 | 33.51 | 0.908 | 0.094 |
| Ex4DGS (Lee et al., 2024) | Retrained (GT Loss) | 32.18 | 0.944 | 0.039 | 35.33 | 0.896 | 0.124 | 36.40 | 0.930 | 0.094 |
| | Distilled (w/o Weight) | 32.39 | 0.945 | 0.041 | 35.44 | 0.896 | 0.126 | 36.37 | 0.930 | 0.095 |
| | Distilled (Ours) | 32.41 | 0.946 | 0.038 | 35.88 | 0.903 | 0.115 | 36.87 | 0.935 | 0.086 |

| Model | Training | Theater | | | Train | | | Average | | |
|---|---|---|---|---|---|---|---|---|---|---|
| | | PSNR | SSIM | LPIPS | PSNR | SSIM | LPIPS | PSNR | SSIM | LPIPS |
| E-D3DGS (Bae et al., 2024) | Retrained (GT Loss) | 30.49 | 0.871 | 0.148 | 30.92 | 0.896 | 0.097 | 32.88 | 0.902 | 0.111 |
| | Distilled (w/o Weight) | 31.73 | 0.878 | 0.150 | 30.85 | 0.898 | 0.099 | 33.26 | 0.907 | 0.110 |
| | Distilled (Ours) | 31.79 | 0.887 | 0.124 | 31.46 | 0.900 | 0.095 | 33.67 | 0.915 | 0.091 |
| STG (Li et al., 2024) | Retrained (GT Loss) | 30.28 | 0.876 | 0.126 | 32.26 | 0.942 | 0.036 | 32.83 | 0.915 | 0.083 |
| | Distilled (w/o Weight) | 31.23 | 0.883 | 0.124 | 32.30 | 0.943 | 0.039 | 33.09 | 0.917 | 0.084 |
| | Distilled (Ours) | 31.35 | 0.883 | 0.124 | 32.20 | 0.943 | 0.038 | 33.11 | 0.917 | 0.082 |
| Ex4DGS (Lee et al., 2024) | Retrained (GT Loss) | 31.85 | 0.886 | 0.122 | 32.11 | 0.934 | 0.050 | 33.57 | 0.918 | 0.086 |
| | Distilled (w/o Weight) | 31.78 | 0.884 | 0.126 | 32.14 | 0.935 | 0.053 | 33.62 | 0.918 | 0.088 |
| | Distilled (Ours) | 32.04 | 0.890 | 0.111 | 32.37 | 0.939 | 0.045 | 33.91 | 0.923 | 0.079 |

## C.2 Qualitative results

Figures 7, 8 present additional qualitative comparisons of MoE-GS across different datasets. MoE-GS effectively routes scene regions to the most suitable experts, resulting in high-fidelity reconstructions that outperform individual models. These results further demonstrate the model's ability to adaptively combine specialized expert outputs for diverse dynamic scenes. In addition, Figure 9 shows distillation qualitative results on the Technicolor dataset.

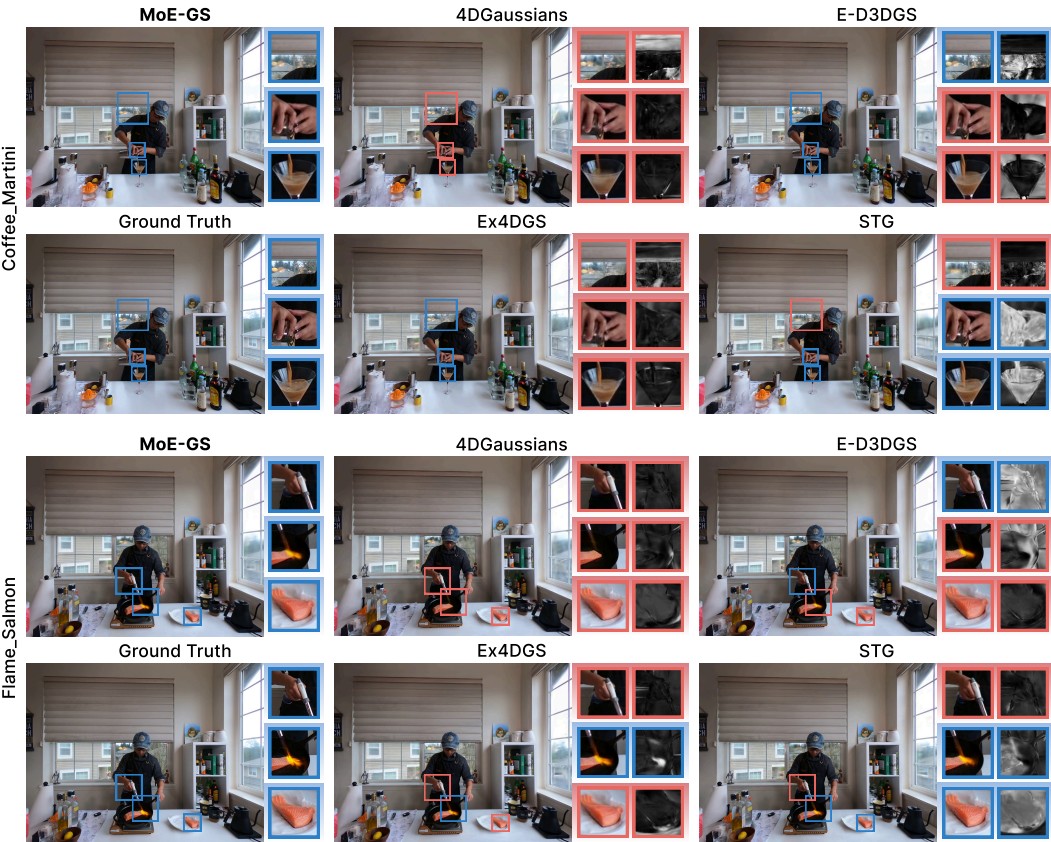

Figure 7: **Additional N3V Qualitative Results.** Comparison of our MoE-GS with other dynamic Gaussian splatting methods on the Neural 3D Video dataset (Li et al., 2022).

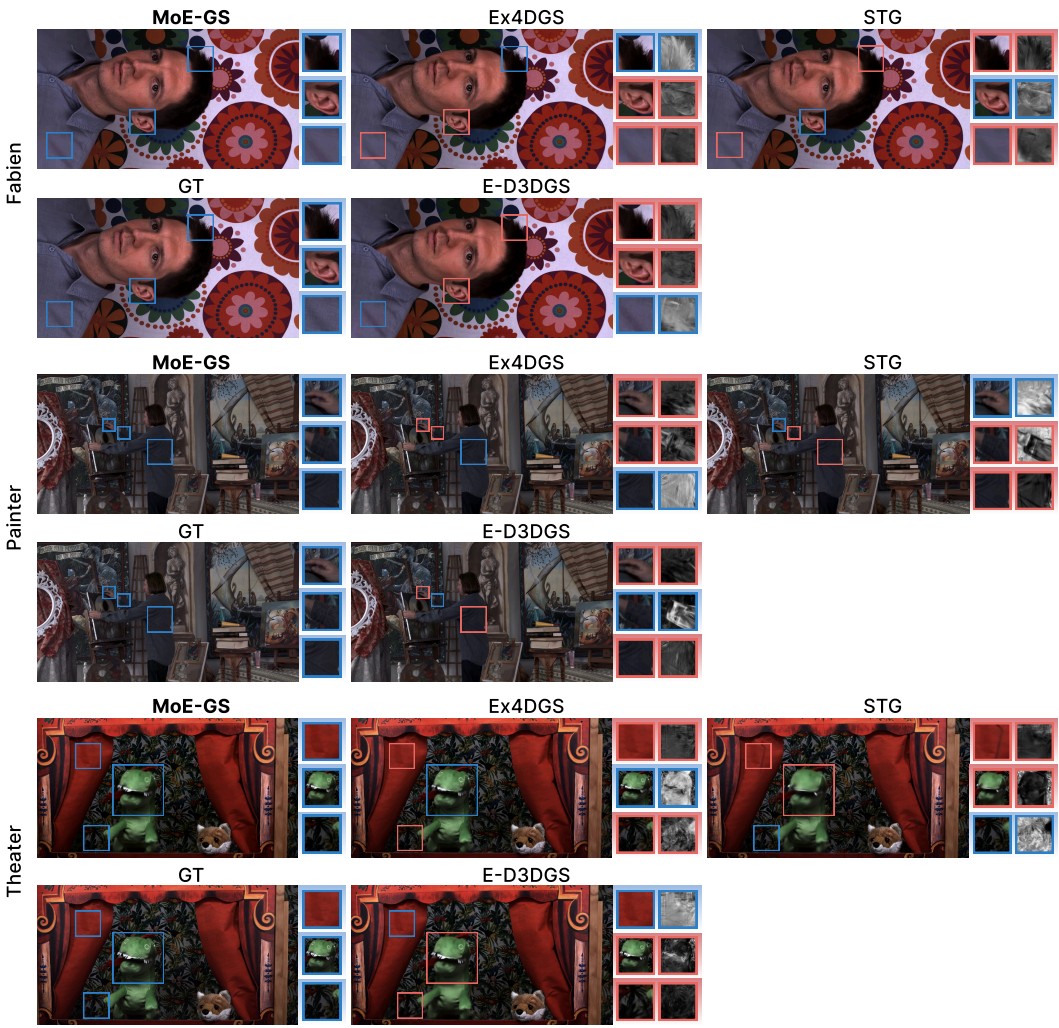

Figure 8: **Additional Qualitative Results on the Technicolor Dataset (Sabater et al., 2017).** Visual comparison of our MoE-GS with other dynamic Gaussian splatting methods.

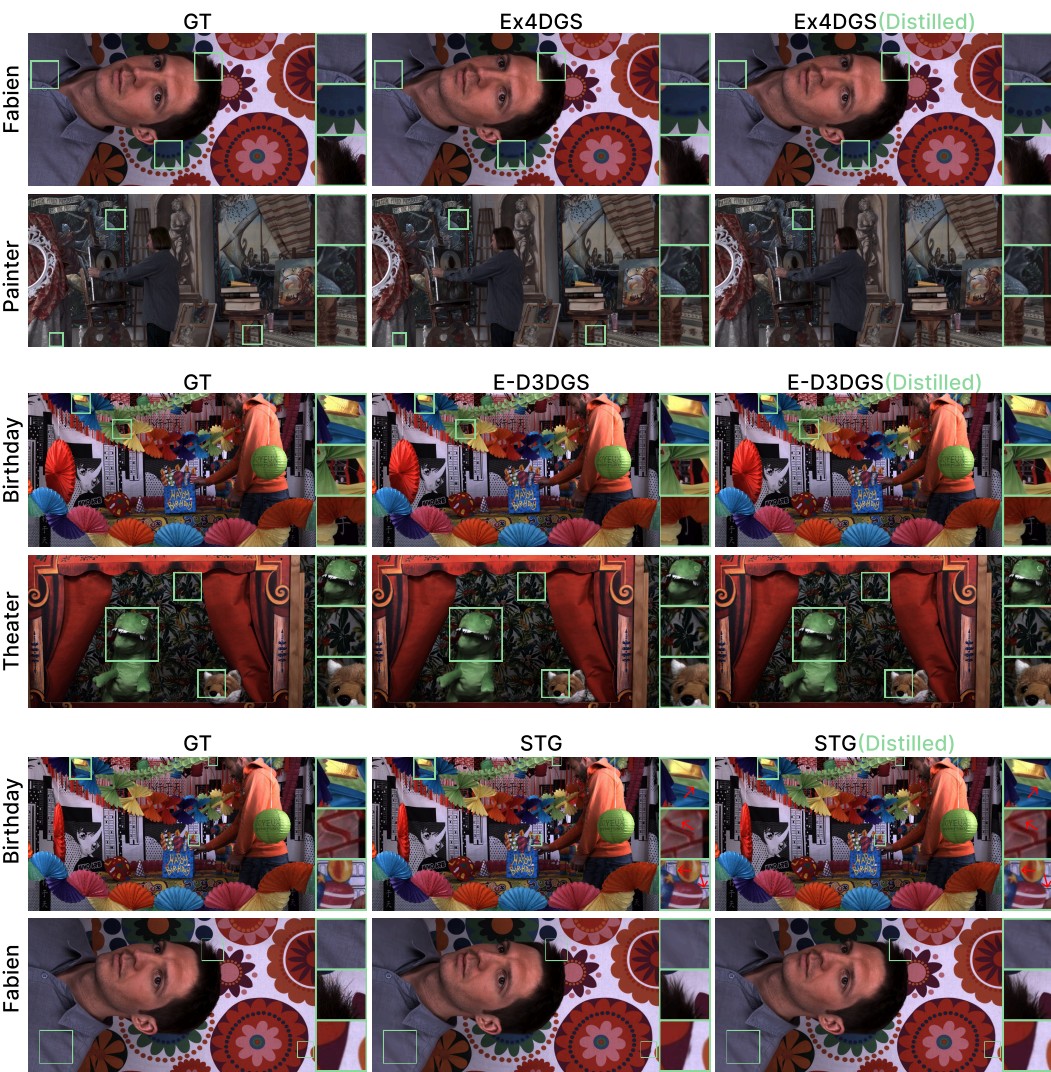

Figure 9: **Qualitative Results of Distillation on the Technicolor Dataset (Sabater et al., 2017).** Visual comparison between retrained and distilled expert models on the Technicolor dataset (Sabater et al., 2017).

# D    GAUSSIAN-LEVEL INTERPRETATION AND GEOMETRY EVALUATION

## D.1    PER-GAUSSIAN CONTRIBUTIONS AND RESPONSIBILITIES

As described in Section 3.2, the router outputs per-pixel expert weights $G'_k(u)$, while each Gaussian $j$ of expert $k$ contributes to pixel $u$ through its volumetric transmittance $T_{k,j}(u)$ and opacity $\alpha_{k,j}(u)$. We first define the raw volumetric contribution of Gaussian $g_{k,j}$ as

$$C_{k,j}(u) = T_{k,j}(u)\,\alpha_{k,j}(u), \tag{12}$$

which measures how much Gaussian $j$ influences the ray at pixel $u$. To lift the pixel-level routing back to the Gaussian domain, we weight this contribution by the router's pixel gate:

$$\tilde{C}_{k,j}(u) = G'_k(u)\,C_{k,j}(u). \tag{13}$$

Because different Gaussians vary in visibility, opacity, and pixel coverage, their raw gated contributions $\tilde{C}_{k,j}$ reflect both geometric scale and routing strength. To isolate the routing effect, we normalize by each Gaussian's total volumetric contribution:

$$\bar{R}_{k,j} = \frac{\sum_{u\in\mathcal{U}} \tilde{C}_{k,j}(u)}{\sum_{u\in\mathcal{U}} C_{k,j}(u)}, \tag{14}$$

where $\mathcal{U}$ denotes the set of all pixel rays from all training views. The resulting $\bar{R}_{k,j}$ measures how strongly Gaussian $j$ is utilized within expert $k$ under the router's pixel-level gates, capturing its effective contribution in the image regions where it is visible. This interpretation arises naturally from our volume-aware router: because Gaussian contributions are computed in 3D before rasterization, pixel-level gating can be consistently traced back to individual Gaussians, enabling a geometry-informed rather than RGB-level interpretation of MoE-GS.

Moreover, different experts operate in heterogeneous deformation spaces, so their Gaussians do not form a one-to-one correspondence in 3D. Pixel space, however, is a shared observation domain across all experts: each Gaussian contributes to the same set of rays through $T_{k,j}(u)\alpha_{k,j}(u)$. The router weights $G'_k(u)$ therefore gate *geometry-conditioned* volumetric contributions, making the lifting operation well-defined and ensuring that the resulting Gaussian responsibilities preserve consistent 3D semantics rather than reflecting any form of 2D blending.

## D.2    POST-HOC GAUSSIAN FUSION USING LIFTING WEIGHTS

Using the normalized responsibilities $\bar{R}_{k,j}$ from Section D.1, we construct a post-hoc unified Gaussian model without any retraining. For each Gaussian $j$ in expert $k$, we keep all geometry attributes—position, scale, rotation, and SH coefficients—unchanged, and adjust only the opacity based on the responsibility weight:

$$\alpha^{\text{fused}}_{k,j} = \bar{R}_{k,j}\,\alpha_{k,j}. \tag{15}$$

The final fused model is obtained by simply concatenating all Gaussians from all experts, using the fused opacities. This unified Gaussian set is rendered with the standard 3D Gaussian Splatting volume renderer—*without* any 2D compositing. The result preserves each expert's geometric structure while modulating its influence according to the router-driven responsibilities, yielding a coherent volumetric representation that reflects geometry-informed routing rather than image-space blending.

## D.3    MULTI-VIEW DEPTH CONSISTENCY EVALUATION

To quantitatively evaluate the geometry of our post-hoc fused Gaussian model, we compute the Multi-view Depth Consistency (MDC), defined as the mean reprojection error between depth maps across all viewpoint pairs at the same timestamp. Given rendered depth maps $\{D^t_i\}$ from viewpoints $\{v_i\}$ at time $t$, we measure consistency over all ordered pairs $(i,j)$ by reprojecting the depth from view $i$ into view $j$ and comparing it against $D^t_j$:

$$\text{MDC} = \frac{1}{|\mathcal{P}|} \sum_{(i,j)\in\mathcal{P}} \left\| \Pi_{v_j}(D^t_i) - D^t_j \right\|_1, \tag{16}$$

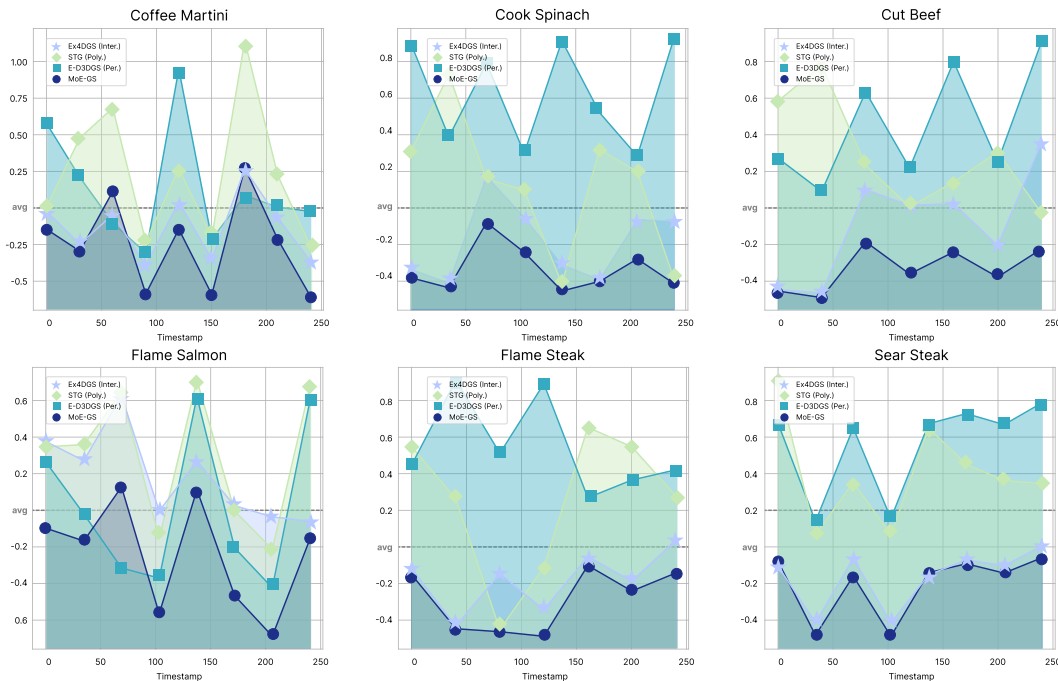

Figure 10: Multi-view Depth Consistency on the N3V dataset. Comparison of our MoE-GS with other dynamic Gaussian Splatting methods (**lower is better**).

where $\mathcal{P}$ denotes the set of all ordered view pairs, and $\Pi_{v_j}(D_i^t)$ is the reprojection of $D_i^t$ into view $j$ using the known camera poses at timestamp $t$.

Fig. 10 reports MDC scores across the entire N3V dataset (Li et al., 2022), comparing our post-hoc fused Gaussian model against all dynamic Gaussian Splatting baselines. Across all scenes and timestamps, MoE-GS consistently achieves lower reprojection error (**lower is better**), indicating improved multi-view geometric stability. This suggests that MoE-GS effectively combines complementary geometric priors from heterogeneous experts, producing a unified Gaussian representation that remains consistent across the sequence rather than relying on appearance-level blending.

These results provide quantitative evidence that the proposed Gaussian-level weighting leads to more coherent underlying geometry. Even though MoE-GS relies on pixel-level gates during training, the fused Gaussian model exhibits globally improved 3D coherence when evaluated using a purely volumetric metric. Averaged over all scenes and timestamps, MoE-GS attains the best MDC performance, demonstrating that our mixture formulation enhances geometric reconstruction quality rather than only photometric fidelity.

# E    ANALYSIS OF EXPERT-SPECIFIC MOTION BEHAVIOR

To better understand the performance variations observed across dynamic Gaussian Splatting methods, we analyze the characteristic motion trajectories produced by each expert (Fig. 11). Although all methods operate on Gaussian primitives, their deformation priors impose fundamentally different inductive biases, leading to distinct behaviors across spatial regions and temporal motion regimes. Below, we summarize the key tendencies of each expert.

**4DGaussians (Wu et al., 2024) (HexPlane canonical deformation).** 4DGaussians produces short, smooth, and highly regular trajectories due to its shared HexPlane canonical representation. The deformation MLP conditions on features interpolated from a global position–time grid, causing spatially adjacent Gaussians to receive nearly identical deformation signals. This results in strong spatial regularization and minimal per-Gaussian variation. While this bias enables stable performance in *static or low-motion regions*, it hinders the representation of *high-speed or rapidly changing motion*, where the canonical features cannot vary sufficiently quickly to encode fine-grained displacement.

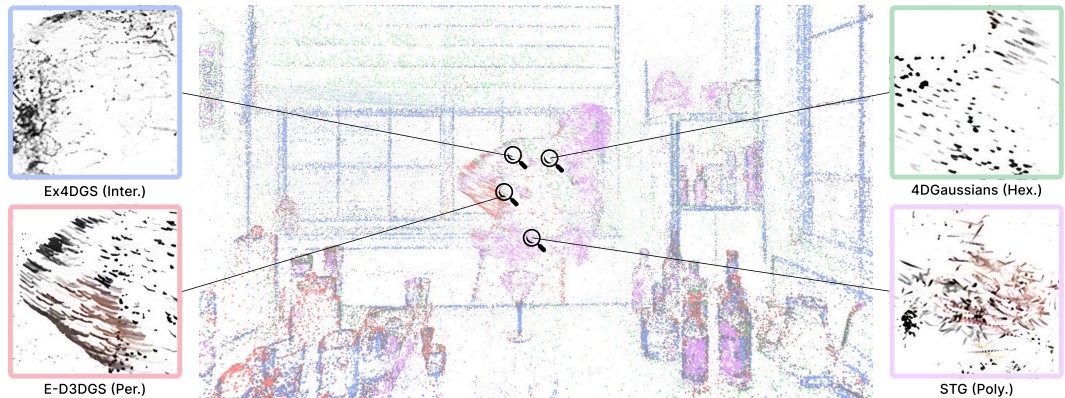

Figure 11: **Motion Trajectory** Comparison across dynamic Gaussian Splatting methods.

Consequently, 4DGaussians trajectories often become overly smooth or collapsed in fast-motion regions.

**E-D3DGS (Bae et al., 2024) (per-Gaussian volumetric deformation).** E-D3DGS exhibits directionally consistent and comparatively higher-velocity trajectories. This behavior arises from two architectural elements: (i) a two-branch deformation network that separates coarse and fine motion, allowing the model to represent *fast and detailed dynamics*; and (ii) a local embedding regularizer encouraging neighboring Gaussians to share similar deformation features, yielding *spatially coherent and locally aligned trajectories*. These effects are further reinforced by the learned per-Gaussian embeddings $z_g$, which act as local motion descriptors. Because the deformation MLP conditions on these embeddings—and nearby Gaussians are regularized to have similar embeddings—the model naturally forms coherent motion clusters with consistent velocity and direction. This embedding-driven structure explains the tightly aligned, high-velocity flows characteristic of E-D3DGS.

**Ex4DGS (Lee et al., 2024) (keyframe interpolation).** Ex4DGS produces motion trajectories that are highly diverse and often "free-form," even among Gaussians that are spatially adjacent. This stems from its interpolation-based formulation: each Gaussian updates its position independently between keyframes, without a learned deformation field or a mechanism enforcing spatial coherence. As a result, nearby Gaussians may move with noticeably different magnitudes or directions. This independence is particularly useful in regions with irregular or rapidly changing motion. By allowing each Gaussian to move independently, Ex4DGS can capture *abrupt, multi-directional displacements* that more structured deformation models tend to oversmooth. However, it also leads to less stable or coordinated trajectories in areas where motion is expected to remain coherent or rigid.

**STG (Li et al., 2024) (polynomial trajectory model).** STG produces globally smooth, low-curvature trajectories due to its use of a low-order polynomial motion parameterization. Unlike Ex4DGS, which relies on independent interpolation, STG learns per-Gaussian polynomial coefficients, providing moderate flexibility while maintaining a strong bias toward smoothly varying motion. Because all Gaussians share the same polynomial motion form— while coefficients are learned individually—STG occupies a middle ground between interpolation-based and deformation-based models. It captures moderate local variation yet enforces spatial alignment across neighboring Gaussians. These inductive biases make STG well suited for *rigid or near-rigid global motion* and *low-frequency temporal changes*, producing trajectories that are directionally consistent with limited curvature.

We emphasize that these motion tendencies are representative rather than absolute, as real-world scenes often contain mixed or ambiguous motion regimes that do not align perfectly with any single deformation prior. Nonetheless, the overall patterns observed across experts explain why no existing dynamic GS method consistently dominates across all spatial or temporal regions. This variability naturally motivates the MoE-GS formulation, which adaptively selects the deformation prior most compatible with the local motion behavior.

Table 12: **Stability across repeated trainings.** Variability exists across runs, but MoE-GS consistently outperforms all single-expert variants.

| Model | PSNR (dB) | | |
|---|---|---|---|
| | Min | Max | Average |
| E-D3DGS (Bae et al., 2024) | 30.78 | 32.33 | 31.19 |
| Ex4DGS (Lee et al., 2024) | 31.43 | 32.10 | 31.78 |
| 4DGaussians (Wu et al., 2024) | 30.18 | 31.43 | 30.70 |
| MoE-GS (N=3) | 32.72 | 33.23 | 33.01 |

# F BEST-OF-N RETRAINING VS. MOE-GS: STRUCTURAL, NOT STATISTICAL, GAINS

To evaluate stability under repeated training, we retrain each individual expert model (E-D3DGS, Ex4DGS, 4DGaussians) five times using identical settings. For MoE-GS, we also train five separate MoE-GS models, each using the corresponding expert set from that run. This ensures that every MoE-GS run is paired with its own independently trained experts, and that both baselines and MoE-GS are compared under identical stochastic conditions.

**(1) Single-expert performance varies across repeated trainings.** As shown in Table 12, dynamic GS methods exhibit noticeable run-to-run variation, a behavior commonly observed in optimization-based reconstruction pipelines. However, even the strongest individual run of any expert does not reach the performance achieved by MoE-GS.

**(2) Expert-specific motion behavior remains qualitatively consistent.** Although absolute scores vary across runs, each expert repeatedly displays the same characteristic motion tendencies described in Appendix E—for example, E-D3DGS tends to produce fast, coherent motion, whereas Ex4DGS more often captures irregular, multi-directional movement. These tendencies appear consistently across repeated trainings, reflecting the deformation priors of each method rather than run-specific randomness.

Taken together, these results indicate that MoE-GS provides a stable improvement beyond what can be obtained by repeated training of any single expert, and that its gains stem from integrating complementary deformation priors rather than from statistical fluctuations between runs.

# G ABLATION ON DISTILLATION WEIGHTING STRATEGIES

Distillation in our framework aims to transfer the complementary reconstruction strengths identified by MoE-GS back into a single expert. We explore whether assigning per-pixel weights to the MoE-guided loss improves this transfer, and compare two strategies: (i) w/o weighting, and (ii) gating-based weighting derived from MoE-GS. All experiments use E-D3DGS (Bae et al., 2024) on the Technicolor dataset.

Table 13: Effect of routing-weighted distillation.

| Method | PSNR (dB) ↑ | SSIM ↑ | LPIPS ↓ |
|---|---|---|---|
| E-D3DGS (Baseline) | 32.88 | 0.902 | 0.111 |
| w/o Weight | 33.26 | 0.907 | 0.110 |
| Routing Weight (Ours) | 33.67 | 0.918 | 0.091 |

Using MoE-derived routing weights yields clear improvements over unweighted distillation (Table 13). These routing weights capture how different experts contribute to each spatial–temporal region, providing supervision signals that encode complementary deformation priors rather than relying on pixel-level error statistics. As a result, the distilled expert benefits from MoE-GS's region-specific strengths and achieves higher reconstruction quality.

