# OpenReview forum: "MoE-GS: Mixture of Experts for Dynamic Gaussian Splatting"
_ICLR.cc/2026/Conference — ICLR 2026 Poster_

### Official Review · Reviewer_eMnv · 2025-10-21

**Soundness:** 2
**Presentation:** 3
**Contribution:** 3
**Rating:** 6
**Confidence:** 5

**Summary:**

This paper MoE-GS employs the concept of MoE in 4D spatial reconstruction, based on the observation that multiple methods hold different individual optimal performances in multiple cases. This pipeline builds upon multiple baseline 4DGS methods and assigns volume-based per-Gaussian router score for each expert. The resulting system achieves SOTA reconstruction quality among baselines. This work also introduces acceleration and distillation techniques to improve the inference efficiency.

**Strengths:**

1. This work introduces MoE into 4D reconstruction, which is a novel solution to enhance the 4D representation capability.
2. The Volume aware Pixel Router serves a core innovative design, providing accurate and effective router score based on the rendering process. This design tackles the difficulty in obtaining a suitable score for GS based MoE, which distinguishes this work from previous MoE structures with direct evaluation score.
3. The visual performances are impressive. Based on the MoE structure which aggregates the strength from multiple baseline methods, the proposed method meticulously achieves significantly high visual quality.
4. The authors cater for the inference efficiency of the MoE model. Although the rendering cost is higher than baselines, it still maintains a real-time rendering.

**Weaknesses:**

1. The primary concern is on the method efficiency. This method requires training multiple baseline 4DGS models along side the MoE router. This design costs significantly high time and GPU resources compared to baseline methods. Besides, the time cost used in deciding the router network is also a concern, i.e., the time spent on training the per-Gaussian dynamics property and router network.
2. The significance of a MoE model in this 4D reconstruction scope. Although the MoE structure is applicable to 4D reconstruction problem, it is not an intuitive solution. The 4D reconstruction requires fitting to each individual scene, while the general LLM MoE requires only one training while reused for any inference. In this case, the cost of training a MoE model is efficient compared to training multiple full LLMs. However, this concept is not versatile in the proposed framework. The method requires full training of the individual method while the MoE-GS aggregates and overfits to each model's specialty. This leads to a conflict that, previous MoE for large models is used to improve the efficiency while maintaining high performance, while this framework takes costs to train MoE and improve the quality. These considerations weaken the necessity of such MoE-GS design. Besides, the 3D/4D reconstruction is random, i.e., multiple runs can yield different results. Does an MoE over multiple runs also improve the overall performance? Are the differences between different baselines large enough for a MoE structure?
3. The hyperparameter in Eq. (11) seems an important hyperparameter for distillation. Its impact on the overall distillation performance is not clearly illustrated.

**Questions:**

1. The time cost should be justified, as the method requires significantly higher resources to maintain the SOTA visual performance.
2. The authors should clarify on their motivations and design choices regarding the MoE structure, as listed in the weakness part. The necessity of such structure should be illustrated. A detailed specialty comparison between the expert models, and why these expert models have these properties  should be provided, which are beneficial to the overall novelty.
3. More ablations are recommended to make the experiments solid.

Based on above considerations, I would like to give a positive recommendation regarding the paper's novelty, and I will adjust my score based on the authors' rebuttal on my concerns.

---

> ### Author Response · Authors · 2025-11-21
> **Comment 1/3 for Reviewer eMnv**
>
> We thank the reviewer for the careful reading and thoughtful comments. Below, we address the reviewer’s questions and provide detailed explanations. The corresponding revisions in the manuscript are marked in blue for clarity. We hope that the responses below, along with the updated manuscript, address the reviewer’s concerns thoroughly. Due to character limits on OpenReview, our response has been divided into three parts for ease of readability. Thank you for your understanding.
>
>
>
> **Weakness 1, Question 1:** The primary concern is on the method efficiency. This method requires training multiple baseline 4DGS models along side the MoE router. This design costs significantly high time and GPU resources compared to baseline methods. Besides, the time cost used in deciding the router network is also a concern, i.e., the time spent on training the per-Gaussian dynamics property and router network.
>
> **Response:**  Thank you for raising this important concern. We agree that MoE-GS uses multiple dynamic GS experts, whereas a standard baseline trains only one. To address this, we have added a detailed training-time analysis to **Sec. 4.2** in the revised manuscript. Importantly, the actual training overhead is substantially lower than what one might expect.
>
> **(1) MoE-GS does not require fully converged experts.**
> Because the router selects the expert whose deformation prior best matches each spatio-temporal region, partially trained experts are often sufficient. To quantify this effect, we train each expert using only a fraction of its usual schedule and evaluate MoE-GS under the reduced-budget setting. As shown in **Sec. 4.2**:
>
> - E-D3DGS requires ~4.2h, STG ~2.2h, and 4DGaussians ~1.5h for full training.
> - Using only 20% of this budget (e.g., 4.2h → ~50 min; 1.5h → ~18 min), MoE-GS still outperforms fully trained single-expert baselines.
>
> This shows that meaningful performance gains are achieved even with substantially reduced expert training time.
>
> **(2) The router overhead is negligible.**
> The router operates on lightweight per-Gaussian parameters and a small MLP. It  accounts for <5% of the total compute and converges rapidly. Thus, the additional cost of MoE-specific components is minimal compared to expert optimization.
>
> Finally, recent feed-forward and generalizable 4D Gaussian Splatting methods (0.04–0.07s per frame) provide promising paths toward eliminating per-scene optimization entirely. MoE-GS can directly adopt such models as experts, which will further reduce training overhead as these architectures mature.
>
> **References:**
>
> [1] Zhang, Kai, et al. "Gs-lrm: Large reconstruction model for 3d gaussian splatting." European Conference on Computer Vision. Cham: Springer Nature Switzerland, 2024.
>
> [2] Wang, Yiming, et al. "Learning Efficient Fuse-and-Refine for Feed-Forward 3D Gaussian Splatting." The Thirty-ninth Annual Conference on Neural Information Processing Systems (2025).

---

> ### Author Response · Authors · 2025-11-21
> **Comment 2/3 for Reviewer eMnv**
>
> **Weakness 2, Question 2:** 	The significance of a MoE model in this 4D reconstruction scope. Although the MoE structure is applicable to 4D reconstruction problem, it is not an intuitive solution. The 4D reconstruction requires fitting to each individual scene, while the general LLM MoE requires only one training while reused for any inference. In this case, the cost of training a MoE model is efficient compared to training multiple full LLMs. However, this concept is not versatile in the proposed framework. The method requires full training of the individual method while the MoE-GS aggregates and overfits to each model's specialty. This leads to a conflict that, previous MoE for large models is used to improve the efficiency while maintaining high performance, while this framework takes costs to train MoE and improve the quality. These considerations weaken the necessity of such MoE-GS design. Besides, the 3D/4D reconstruction is random, i.e., multiple runs can yield different results. Does an MoE over multiple runs also improve the overall performance? Are the differences between different baselines large enough for a MoE structure?
>
>
> **Response:**  Thank you for the thoughtful questions. We agree that MoE in LLMs and MoE-GS address fundamentally different goals: LLM MoEs aim for inference-time sparsity, while dynamic 4D reconstruction requires per-scene optimization. In our setting, the motivation for MoE is not efficiency but the need to handle multiple, co-existing motion regimes that cannot be represented faithfully by any single deformation prior.
>
> (1) **Why MoE is necessary in 4D reconstruction.**
> Dynamic Gaussian Splatting methods differ primarily through their *deformation priors*, which impose distinct inductive biases on motion:
> - **E-D3DGS** → fast, locally coherent motion
> - **Ex4DGS** → irregular or multi-directional motion
> - **STG** → globally smooth trajectories
> - **4DGS** → static or low-motion backgrounds
>
> As shown in our motion-analysis examples in **Fig. 1 ([project page](https://moegs-moe-gs.static.hf.space/index.html#analysis))** and **Appendix F**, these behaviors arise from each model’s deformation formulation rather than random training variation. In practice, each expert excels in the motion regime aligned with its inductive bias, making it unlikely for a single model to cover all regimes simultaneously. MoE-GS addresses this limitation by routing each spatio–temporal region to the expert best suited to its underlying motion, leveraging complementary strengths in a unified representation.
>
> (2) **MoE-GS does not require fully trained experts.**
> Although multiple experts are used, MoE-GS does *not* rely on fully converged models. As shown in **Sec. 4.2**, when experts are trained with only 20% of their usual budget, MoE-GS still outperforms fully trained single-expert baselines. The router effectively extracts complementary signals even from partially optimized experts, making the practical overhead far smaller than full multi-model training.
>
> (3) **MoE-GS does not simply aggregate expert idiosyncrasies.**
> We provide a Gaussian-level analysis in **Appendix E** showing that routing produces coherent 3D structure, including:
> - clearer density distributions,
> - more stable occlusion boundaries,
> - fewer temporal inconsistencies.
>
> These improvements remain even after post-hoc opacity fusion using the router’s per-Gaussian responsibilities, indicating that MoE-GS forms a genuine 3D-consistent mixture rather than inheriting noise from individual experts.
>
> (4) **Regarding randomness: MoE-GS improves both median accuracy and stability.**
> As shown in **Appendix G**, dynamic GS models exhibit run-to-run variation. However, across repeated runs, MoE-GS consistently achieves:
> - higher median performance, and
> - lower variance.
>
> This occurs because routing aggregates the structured strengths of experts while smoothing out run-specific noise, leading to more stable results than any individual expert.
>
> We have incorporated these explanations—motivation, expert specialization, synergy, and run-to-run robustness—into **Sec. 1, Sec. 4.2, and Appendix F–G** of the revised manuscript.
>
> **References:**
>
> [1] Bae, Jeongmin, et al. "Per-gaussian embedding-based deformation for deformable 3d gaussian splatting." European Conference on Computer Vision. Cham: Springer Nature Switzerland, 2024.
>
> [2] Lee, Junoh, et al. "Fully explicit dynamic gaussian splatting." Advances in Neural Information Processing Systems 37 (2024): 5384-5409.
>
> [3] Li, Zhan, et al. "Spacetime gaussian feature splatting for real-time dynamic view synthesis." Proceedings of the IEEE/CVF Conference on Computer Vision and Pattern Recognition. 2024.
>
> [4] Wu, Guanjun, et al. "4d gaussian splatting for real-time dynamic scene rendering." Proceedings of the IEEE/CVF conference on computer vision and pattern recognition. 2024.
>
> ---

---

> > ### Author Response · Authors · 2025-11-21
> > **Comment 3/3 for Reviewer eMnv**
> >
> > **Weakness 3:** 	The hyperparameter in Eq. (11) seems an important hyperparameter for distillation. Its impact on the overall distillation performance is not clearly illustrated.
> >
> >
> > **Response:** Thank you for pointing this out. The coefficient $\lambda$ in Eq. (11) determines the relative strength of the MoE-guided distillation term with respect to the standard ground-truth reconstruction loss. When $\lambda$ is small, the MoE-guided term becomes more influential, while a large $\lambda$ reduces the effect of distillation and makes the student behave more similarly to its original baseline.
> >
> > Empirically, we find that the distillation performance is stable within a moderate range of $\lambda$ (0.1–0.3). Within this interval, the ground-truth supervision remains the primary signal, while the MoE-guided term provides a useful refinement without noticeably altering the optimization dynamics. Performance differences across this range remain within 0.1–0.15 dB. Therefore, we fix $\lambda = 0.2$ in all experiments as a balanced choice that retains the original reconstruction quality while modestly incorporating the spatial–temporal guidance provided by MoE-GS.
> >
> > ---
> >
> > **Question 3:**  More ablations are recommended to make the experiments solid.
> >
> > **Response:** Thank you for the suggestion. In the revised manuscript, we have added several new ablations and diagnostic studies to strengthen the empirical analysis:
> >
> > 1. **Training-budget ablation (Sec. 4.2)** :
> > We evaluate MoE-GS when experts are trained with only 50%, 20%, and 10% of their original budget. MoE-GS continues to outperform fully trained single experts, showing that it does not rely on fully converged experts.
> >
> > 2. **Geometry-level evaluation (Appendix E)**:
> > We compute multi-view depth consistency to examine geometric coherence beyond photometric metrics. MoE–GS achieves the best MDC across all N3V scenes.
> >
> > 3. **Expert motion–behavior analysis (Appendix F)**:
> > We analyze the characteristic motion regimes each expert specializes in, providing causal insight into the complementary strengths revealed in Fig. 1.
> >
> > 4. **Multi-run stability analysis (Appendix G)**:
> > Because dynamic GS optimization can yield slightly different results across runs, we additionally perform repeated training runs for both individual experts and MoE-GS. MoE-GS consistently achieves higher median performance and lower variance across runs, demonstrating that routing reduces optimization noise rather than amplifying it.
> >
> > 5. **Distillation weighting–strategy ablation (Appendix H)**:
> > We compare three strategies for weighting the distillation loss—no weighting, and MoE routing–weight weighting. Routing-based weights yield the highest reconstruction quality, showing that the MoE-derived supervision provides beneficial spatial–temporal guidance.
> >
> > These additional ablations together offer a more complete and robust empirical picture of MoE-GS. We thank the reviewer for the suggestion.

---

> ### Comment · Reviewer_eMnv · 2025-11-26
> **Thanks for the rebuttal.**
>
> I appreciate the authors’ detailed rebuttal. I have carefully re-examined the rebuttal content along with the updated manuscript. The responses adequately address most of my previous concerns, and the overall quality of the paper has been improved. While the authors provide extensive experiments demonstrating that the proposed method achieves state-of-the-art performance with partially converged expert models and comparable training time, the method still introduces non-negligible complexity during practical usage. This is not necessarily a drawback, but rather an inherent characteristic of the proposed approach. This concern regarding efficiency and complexity does not diminish the novelty and significance of the work. Considering its contributions, I would like to maintain my positive score.

---

### Official Review · Reviewer_AHQB · 2025-10-31

**Soundness:** 4
**Presentation:** 2
**Contribution:** 3
**Rating:** 6
**Confidence:** 3

**Summary:**

This paper presents MoE-GS, the first approach to apply a Mixture-of-Experts (MoE) architecture to Dynamic Gaussian Representation. Using a Volume-aware Pixel Router, MoE-GS mixes the Gaussians produced by experts specialized in representing specific Gaussians within the same scene. Although this introduces runtime overhead, the authors mitigate it with single-pass multi-expert rendering and gate-aware Gaussian pruning. As a result, experiments on the N3V and Technicolor datasets show that MoE-GS achieves better view-synthesis quality and efficiency than prior state-of-the-art methods.

**Strengths:**

	MoE-GS is the first work to introduce a Mixture-of-Experts architecture to dynamic Gaussian splatting, offering a new solution to the problem that a single model cannot handle scene-specific diversity.

	The proposed MoE-GS model consistently achieves higher rendering quality than existing SOTA methods on complex dynamic scenes and demonstrated its superiority across diverse datasets (N3V, Technicolor).

	The authors acknowledge the drawbacks of the MoE architecture, namely increased computational load and reduced speed and jointly propose techniques to mitigate these issues.

**Weaknesses:**

Major weaknesses are as below:

	In LLMs, MoE is typically used at inference time to reduce the number of activated parameters [1]. By contrast, this paper uses MoE as a mechanism for mixing multiple models. Nevertheless, it suggests that as the number of models increases, computational resource demands grow, while the performance gains are unlikely (at a fine-grained level, e.g. a patch of novel view image) to surpass the best performance of any single model.

[1] Dai et al. “DeepSeekMoE: Towards Ultimate Expert Specialization in Mixture-of-Experts Language Models.” CoRR. 2024.

	The rationale for using the variational models (4DGaussians (Wu et al., 2024), E-D3DGS (Bae et al., 2024), Ex4DGS (Lee et al., 2024), and STG (Li et al., 2024)) is sufficient. However, the paper lacks analysis of which aspects each model is strong in and why. It also lacks an examination of whether there are synergies among the models and what performance gains such synergies bring.

Minor comments or improvements are as below:

	To emphasize that MoE-GS performs well across all aspects, including MoE-GS’s results in Figure 1 would help clarify that it achieves higher performance than all other methods.

	Do all models require training? If so, reporting the training time for each model and providing an analysis of the increased training time would be helpful.

I will reconsider the score when all the concerns are handled well.

**Questions:**

	In Equation (6), what is the rationale for multiplying the time variable by the time-dependent per-Gaussian weights?

	In the Theater results of Table 2, MoE-GS underperforms relative to HyperReel. I hypothesize that this is because HyperReel is not employed as an expert; would incorporating HyperReel as an expert improve performance?

---

> ### Author Response · Authors · 2025-11-21
> **Comment 1/3 for Reviewer AHQB**
>
> We thank the reviewer for the careful reading and thoughtful comments. Below, we address the reviewer’s questions and provide detailed explanations. The corresponding revisions in the manuscript are marked in blue for clarity. We hope that the responses below, along with the updated manuscript, address the reviewer’s concerns thoroughly. Due to character limits on OpenReview, our response has been divided into three parts for ease of readability. Thank you for your understanding.
>
>
>
> **Weakness 1:** In LLMs, MoE is typically used at inference time to reduce the number of activated parameters [1]. By contrast, this paper uses MoE as a mechanism for mixing multiple models. Nevertheless, it suggests that as the number of models increases, computational resource demands grow, while the performance gains are unlikely (at a fine-grained level, e.g. a patch of novel view image) to surpass the best performance of any single model.
>
> **Response:** Thank you for the insightful question. We agree that MoE usage in LLMs is largely driven by inference-time sparsity. Dynamic 4D scene reconstruction, however, involves a very different limitation: a single Gaussian Splatting model is aligned with only one type of deformation behavior, while real scenes often contain multiple, co-existing motion regimes that a single inductive bias cannot represent faithfully.
>
> Each dynamic GS method used as an expert in our framework operates in a distinct 3D deformation space—canonical HexPlane deformation, per-Gaussian volumetric deformation, polynomial trajectories, or interpolation-based motion. These differences are structural rather than architectural: they cause models to specialize in fundamentally different motion patterns. As shown in our updated motion analysis (**[project page](https://moegs-moe-gs.static.hf.space/index.html#analysis)**), some experts consistently handle fast, locally structured motion well, others specialize in irregular multi-directional motion, and others in smooth global trajectories. This specialization arises from their deformation priors rather than from training noise, and cannot be replicated by a single model. A detailed explanation of these expert-specific motion behaviors is provided in **Appendix F**.
>
> The role of MoE-GS is therefore not to surpass the best expert in every local patch, but to provide a globally coherent 4D representation by routing each region of the scene to the deformation prior that is most compatible with its underlying dynamics. This effect is difficult to observe from photometric metrics alone, so we include two additional analyses:
> - Post-hoc blended GS, rendered entirely with the standard volumetric 3DGS pipeline, which shows improved density coherence, occlusion stability, and reduced ghosting
> - Multi-view Depth Consistency (MDC), where MoE-GS achieves the lowest depth reprojection error across all N3V scenes.
>
> Together, these results show that MoE-GS improves geometric and temporal consistency, qualities that no single expert achieves uniformly. Detailed results for both analyses are provided in **Appendix E**.
>
> Thus, while MoE-GS is not designed for computational sparsity as in LLM MoEs, the underlying rationale is analogous: different experts capture complementary motion and deformation behaviors, and leveraging these complementary priors yields a representation unattainable by any single model.
>
> **References:**
>
> [1] Dai et al. “DeepSeekMoE: Towards Ultimate Expert Specialization in Mixture-of-Experts Language Models.” CoRR. 2024.
>
> [2] Wu, Guanjun, et al. "4d gaussian splatting for real-time dynamic scene rendering." Proceedings of the IEEE/CVF conference on computer vision and pattern recognition. 2024.
>
> [3] Bae, Jeongmin, et al. "Per-gaussian embedding-based deformation for deformable 3d gaussian splatting." European Conference on Computer Vision. Cham: Springer Nature Switzerland, 2024.
>
> [4] Li, Zhan, et al. "Spacetime gaussian feature splatting for real-time dynamic view synthesis." Proceedings of the IEEE/CVF Conference on Computer Vision and Pattern Recognition. 2024.
>
> [5] Lee, Junoh, et al. "Fully explicit dynamic gaussian splatting." Advances in Neural Information Processing Systems 37 (2024): 5384-5409.
>
> ---

---

> ### Author Response · Authors · 2025-11-21
> **Comment 2/3 for Reviewer AHQB**
>
> **Weakness 2:** The rationale for using the variational models (4DGaussians (Wu et al., 2024), E-D3DGS (Bae et al., 2024), Ex4DGS (Lee et al., 2024), and STG (Li et al., 2024)) is sufficient. However, the paper lacks analysis of which aspects each model is strong in and why. It also lacks an examination of whether there are synergies among the models and what performance gains such synergies bring.
>
> **Response:**  Thank you for the insightful suggestion. We agree that clarifying why each variational model excels in different scenarios—and how their strengths complement one another—is essential for understanding the motivation behind MoE-GS. In the revision, we provide a clearer causal explanation grounded in the deformation priors of each method, and we update **Fig. 1** and **[project page](https://moegs-moe-gs.static.hf.space/index.html#analysis)** using the new motion-level diagnostics introduced in **Appendix F**.
>
> Each dynamic GS method operates in a distinct deformation space, which directly shapes its motion behavior:
>
> - **4DGaussians (Wu et al., 2024).**
> Its HexPlane-based canonical deformation imposes strong spatial regularization, producing short, smooth, and highly consistent trajectories. This bias leads to reliable performance in static or low-motion background regions.
>
> - **E-D3DGS (Bae et al., 2024).**
> Its deformation network separates slow and fast motion through two MLP branches, and a local embedding regularizer encourages neighboring Gaussians to share similar motion directions. As shown in our trajectory statistics, this yields locally coherent yet high-frequency motion, making E-D3DGS effective in scenes with fast but structured dynamics.
>
> - **Ex4DGS (Lee et al., 2024).**
> Independent keyframe interpolation causes Gaussians to follow free-form, multi-directional trajectories, which naturally aligns with abrupt or irregular motion regimes.
>
> - **STG (Li et al., 2024).**
> Its low-order polynomial trajectory parameterization produces globally smooth, low-curvature motion, making STG well suited for rigid or near-rigid global movement.
>
> These deformation priors provide a mechanistic explanation for the specialization trends observed across scenes: different motion regimes align with different inductive biases, so no single model can dominate across all spatial or temporal regions—a trend consistently observed in **Fig. 1**.
>
> - **Synergy among experts.**
> Because each expert captures a distinct subset of motion behaviors, their strengths are complementary rather than redundant. The router in MoE-GS consistently assigns regions with different motion structures to different experts. This specialization produces genuine synergy: by leveraging heterogeneous deformation priors, MoE-GS achieves more stable and coherent reconstruction than any individual expert.
>
> **References:**
>
> [1] Wu, Guanjun, et al. "4d gaussian splatting for real-time dynamic scene rendering." Proceedings of the IEEE/CVF conference on computer vision and pattern recognition. 2024.
>
> [2] Bae, Jeongmin, et al. "Per-gaussian embedding-based deformation for deformable 3d gaussian splatting." European Conference on Computer Vision. Cham: Springer Nature Switzerland, 2024.
>
> [3] Li, Zhan, et al. "Spacetime gaussian feature splatting for real-time dynamic view synthesis." Proceedings of the IEEE/CVF Conference on Computer Vision and Pattern Recognition. 2024.
>
> [4] Lee, Junoh, et al. "Fully explicit dynamic gaussian splatting." Advances in Neural Information Processing Systems 37 (2024): 5384-5409.
>
> ---
>
> **Weakness 3:** To emphasize that MoE-GS performs well across all aspects, including MoE-GS’s results in Figure 1 would help clarify that it achieves higher performance than all other methods.
>
> **Response:** Thank you for the helpful suggestion. We have reflected this in the revision and added the MoE-GS results to **Fig. 1**. We appreciate the reviewer’s comment.
>
> ---

---

> > ### Author Response · Authors · 2025-11-21
> > **Comment 3/3 for Reviewer AHQB**
> >
> > **Weakness 4:** Do all models require training? If so, reporting the training time for each model and providing an analysis of the increased training time would be helpful.
> >
> > **Response:**  Thank you for raising this point. Yes—each expert model in MoE-GS is trained, as dynamic Gaussian Splatting methods are optimization-based rather than feed-forward. In our two-stage pipeline, Stage 1 trains each expert following its original procedure, and Stage 2 optimizes only the lightweight router, which adds negligible overhead (<5% of total compute).
> >
> > To address the reviewer’s request, we report the wall-clock training times for all experts in the revised manuscript (**Sec. 4.2**). In our implementation, training takes approximately 4.2 hours for E-D3DGS, 2.2 hours for STG, and 1.5 hours for 4DGaussians. We additionally include a reduced-budget experiment showing that MoE-GS remains highly effective even when each expert is trained with only **20%** of its usual schedule (e.g., reducing a 4.2-hour expert to ~50 minutes). Under this reduced-budget setting, MoE-GS still surpasses **fully trained** single-expert baselines, indicating that fully converged experts are not required and that the practical training overhead is significantly smaller than the theoretical N× factor.
> >
> > Finally, recent progress in **feed-forward 3D/4D Gaussian Splatting** has demonstrated that dynamic Gaussians can be generated at near real-time rates (0.04–0.07s per frame). As these architectures mature, they can be incorporated directly as experts in MoE-GS, further reducing training time while preserving the same routing formulation.
> >
> > **References:**
> >
> > [1] Zhang, Kai, et al. "Gs-lrm: Large reconstruction model for 3d gaussian splatting." European Conference on Computer Vision. Cham: Springer Nature Switzerland, 2024.
> >
> > [2] Wang, Yiming, et al. "Learning Efficient Fuse-and-Refine for Feed-Forward 3D Gaussian Splatting." The Thirty-ninth Annual Conference on Neural Information Processing Systems (2025).
> >
> >
> > ---
> >
> > **Question 1:** In Equation (6), what is the rationale for multiplying the time variable by the time-dependent per-Gaussian weights?
> >
> > **Response:** Thank you for the question. In Eq. (6), the time multiplication serves as a normalization that allows the per-Gaussian time-dependent weight to encode the relative temporal influence of a Gaussian. Different Gaussians may be active over different temporal ranges or exhibit different motion magnitudes; using  $w_i^{\text{time}}$ alone would give unbalanced contributions across Gaussians. Multiplying by the normalized time variable stabilizes the scale of this term and ensures that the router interprets it as a temporal modulation factor rather than an absolute offset.
> >
> > ---
> >
> > **Question 2:** In the Theater results of Table 2, MoE-GS underperforms relative to HyperReel. I hypothesize that this is because HyperReel is not employed as an expert; would incorporating HyperReel as an expert improve performance?
> >
> > **Response:**  Thank you for the question. HyperReel indeed performs well in Theater scenarios. However, HyperReel is based on a radiance-field formulation rather than a Gaussian-based deformation model, and the two representations differ fundamentally in both rendering and optimization. For this reason, HyperReel cannot be directly used as an expert within our Gaussian-based MoE framework. Our focus in this work is on experts that share a compatible Gaussian representation, allowing stable routing and training.
> >
> > We appreciate the reviewer’s insightful observation, and we believe extending MoE-GS to incorporate non-Gaussian experts—such as radiance-field models—would be an exciting direction for future work that could further broaden the applicability of the framework.

---

> ### Comment · Reviewer_AHQB · 2025-11-28
>
> Thank you to the authors for their clear responses. Most of my questions have been resolved, and they have incorporated my suggestions.
>
> The updated paper and appendix further reinforce the validity of the method proposed by the authors. The authors have clarified the scope of applying MoE, which, although different from the usual purpose of MoE in LLMs, proposes a novel way of ensembling various 4DGS methods and demonstrates the effectiveness of this approach. With this method, each expert model operates in the regions where it is specialized, making it a good way to integrate them. I will maintain my positive score.

---

### Official Review · Reviewer_bdci · 2025-10-31

**Soundness:** 2
**Presentation:** 2
**Contribution:** 2
**Rating:** 2
**Confidence:** 4

**Summary:**

This paper introduces MoE-GS (Mixture of Experts for Dynamic Gaussian Splatting), the first framework to apply mixture-of-experts techniques to dynamic 3D scene reconstruction using Gaussian splatting. The authors observe that existing dynamic Gaussian splatting methods show inconsistent performance across different scenes, spatial regions, and temporal frames, indicating no single approach handles all dynamic challenges effectively. To address this, MoE-GS integrates multiple specialized expert models through a novel Volume-aware Pixel Router that adaptively blends expert outputs by projecting Gaussian-level weights into pixel space via differentiable splatting. The framework also includes efficiency optimizations like single-pass multi-expert rendering and gate-aware Gaussian pruning, plus a knowledge distillation strategy to transfer MoE performance to individual experts for lightweight deployment. Experiments on the N3V and Technicolor datasets demonstrate that MoE-GS consistently outperforms state-of-the-art methods in rendering quality while maintaining practical efficiency.

**Strengths:**

1. The paper provides a solid diagnostic study (Fig. 1) showing how different dynamic Gaussian Splatting models reach performance peaks at different scenes, spatial regions, and time steps. This empirical observation is valuable to the community and highlights an important open problem of model generalization in dynamic reconstruction.

2. Even if the current MoE-GS realization focuses on image-level blending, the paper raises the broader idea of combining specialized dynamic reconstruction models adaptively. This perspective could motivate subsequent studies toward true 3D-level mixtures or unified dynamic representations.

**Weaknesses:**

1. Has conceptual mismatch between its stated motivation and its technical realization. In 3D reconstruction, **novel view synthesis is merely a means to validate that the reconstructed geometry is correct, not the ultimate goal itself**. However, MoE-GS operates entirely at the image level, where the proposed Volume-aware Pixel Router blends rendered 2D outputs from multiple experts using per-pixel softmax weighting. Each expert maintains an independent 3D Gaussian representation, and **the router does not modify or unify these geometries**. Consequently, the method **only improves rendering appearance (e.g., PSNR, SSIM)** through image-space ensembling but does not enhance or reconcile the underlying 3D geometry. The core mechanism of MoE-GS is effectively **no different from mixing several rendered images with learned blending weights**, since the entire representation being optimized is the final rendered image rather than the underlying 3D structure, with no merged model output. Consequently, the claimed benefit for “dynamic reconstruction” is not real.

While I understand that per-Gaussian routing may be technically challenging, I would appreciate clarification on whether the authors believe their Mixture-of-Experts design can genuinely improve geometric reconstruction quality rather than primarily enhancing rendering appearance.

2. In figure 1, the paper convincingly demonstrates empirically that different dynamic Gaussian Splatting (GS) methods achieve different performance peaks across scenes, spatial regions, and time steps, but it does not provide a clear causal explanation for why these variations occur. Adding deeper theoretical or empirical insight into why existing GS methods differ in scene-, region-, and time-dependent performance will increase interpretability and scientific understanding of the proposed approach’s foundations.

**Questions:**

Please refer to the Weaknesses above.

---

> ### Author Response · Authors · 2025-11-21
> **Comment 1/3 for Reviewer bdci**
>
> We thank the reviewer for the careful reading and thoughtful comments. Below, we address the reviewer’s questions and provide detailed explanations. The corresponding revisions in the manuscript are marked in blue for clarity. We hope that the responses below, along with the updated manuscript, address the reviewer’s concerns thoroughly. Due to character limits on OpenReview, our response has been divided into three parts for ease of readability. Thank you for your understanding.
>
>
> **Weakness 1:** Has conceptual mismatch between its stated motivation and its technical realization. In 3D reconstruction, novel view synthesis is merely a means to validate that the reconstructed geometry is correct, not the ultimate goal itself. However, MoE-GS operates entirely at the image level, where the proposed Volume-aware Pixel Router blends rendered 2D outputs from multiple experts using per-pixel softmax weighting. Each expert maintains an independent 3D Gaussian representation, and the router does not modify or unify these geometries. Consequently, the method only improves rendering appearance (e.g., PSNR, SSIM) through image-space ensembling but does not enhance or reconcile the underlying 3D geometry. The core mechanism of MoE-GS is effectively no different from mixing several rendered images with learned blending weights, since the entire representation being optimized is the final rendered image rather than the underlying 3D structure, with no merged model output. Consequently, the claimed benefit for “dynamic reconstruction” is not real. While I understand that per-Gaussian routing may be technically challenging, I would appreciate clarification on whether the authors believe their Mixture-of-Experts design can genuinely improve geometric reconstruction quality rather than primarily enhancing rendering appearance.
>
> **Response:**
> Thank you for raising this important conceptual concern. We agree that MoE-GS does not explicitly fuse or modify the 3D Gaussian parameters of the individual experts, and we clarified this more explicitly in the revision.
>
> Although the router weight $G’_k(u)$ is applied after per-expert volumetric accumulation, the quantities being combined are *geometry-conditioned* 3D renderings rather than raw RGB values. In our revision, we expand **Section 3.2** and provide a detailed Gaussian-level interpretation in **Appendix E**, showing that pixel-level gates can be consistently lifted back to individual Gaussians through their volumetric contributions.
>
> To summarize the key idea, each Gaussian $j$ in expert $k$ contributes to a pixel ray via
>
> $C_{k,j}(u)=T_{k,j}(u)\cdot \alpha_{k,j}(u),$
>
> and the router modulates this *volumetric* quantity, not color values:
>
> $\tilde{C}\_{k,j}(u)=G'\_{k}(u)\cdot C\_{k,j}(u).$
>
> Aggregating $\tilde{C}_{k,j}$ across all rays yields a normalized responsibility
>
> $\bar{R}\_{k,j}=
> \frac{\sum\_{u\in\mathcal{U}}\tilde{C}\_{k,j}(u)}
>      {\sum\_{u\in\mathcal{U}}C\_{k,j}(u)},$
>
> which reflects how strongly each Gaussian is selected by the router across the sequence. This lifting is well-defined because experts operate in heterogeneous deformation spaces with no one-to-one correspondence in 3D, whereas pixel space is the shared observation domain. Thus, routing behavior can be mapped back to the Gaussian domain only through visibility-weighted volumetric contributions, not through 2D blending.
>
> Using these responsibilities, we construct a post-hoc fused Gaussian model by reweighting only the opacity of each Gaussian, without retraining or modifying geometry:
>
> $\alpha^{\mathrm{fused}}\_{k,j}=\bar{R}\_{k,j}\cdot \alpha\_{k,j}.$
>
> This fused model is rendered using the **standard 3D Gaussian Splatting volumetric renderer**, with no 2D compositing. Despite this minimal change, the fused representation exhibits cleaner density, fewer ghosting artifacts, and more stable occlusion boundaries than any individual expert. Spiral-view visualizations (**[project page](https://moegs-moe-gs.static.hf.space/index.html#spiral)**) further reveal a coherent 3D occupancy structure emerging from the mixture.

---

> ### Author Response · Authors · 2025-11-21
> **Comment 2/3 for Reviewer bdci**
>
> We additionally evaluate Multi-view Depth Consistency (MDC)—a geometry-level measure of cross-view depth coherence. Across all scenes in the N3V dataset, the fused model achieves the lowest reprojection error, demonstrating that MoE-GS yields improved geometric consistency, not merely higher PSNR or SSIM.
>
> Taken together, these findings show that although MoE-GS does not fuse geometry during training, its volumetric-responsibility formulation naturally enables a well-defined post-hoc unification of heterogeneous experts. The fused Gaussian model, combined with MDC improvements and 3D visualizations, demonstrates that MoE-GS forms a coherent 3D occupancy structure that single-expert methods cannot express. In this sense, MoE-GS not only improves geometric consistency but also provides a practical path toward **explicit geometry fusion** in dynamic GS settings.
>
> We sincerely appreciate the reviewer’s insightful comments, which helped us articulate these geometric properties more clearly. The feedback directly strengthened the conceptual framing of our method and led to the explicit analysis and post-hoc fusion results included in the revised manuscript.
>
> **References:**
>
> [1] Li, Tianye, et al. "Neural 3d video synthesis from multi-view video." Proceedings of the IEEE/CVF conference on computer vision and pattern recognition. 2022.

---

> ### Author Response · Authors · 2025-11-21
> **Comment 3/3 for Reviewer bdci**
>
> **Weakness 2:** In figure 1, the paper convincingly demonstrates empirically that different dynamic Gaussian Splatting (GS) methods achieve different performance peaks across scenes, spatial regions, and time steps, but it does not provide a clear causal explanation for why these variations occur. Adding deeper theoretical or empirical insight into why existing GS methods differ in scene-, region-, and time-dependent performance will increase interpretability and scientific understanding of the proposed approach’s foundations.
>
>
> **Response:**
> Thank you for raising this important point. We agree that Fig. 1 benefits from a clearer causal explanation of why dynamic GS methods exhibit different performance peaks. In the revision, we provide a motion-level diagnostic across experts in **[project page](https://moegs-moe-gs.static.hf.space/index.html#analysis)** and update **Fig. 1** to reflect these findings. This analysis shows that the specialization patterns arise directly from the deformation priors encoded in each model.
>
> - **4DGS (HexPlane canonical deformation).**
> 4DGS relies on a shared HexPlane canonical representation, which imposes strong spatial regularization and results in smooth, low-variation deformation. As a consequence, it performs reliably in static or low-motion background regions, but cannot capture high-speed or rapidly changing motion.
>
> - **E-D3DGS (per-Gaussian volumetric deformation).**
> E-D3DGS exhibits a different type of specialization.
> (i) Its deformation network factors slow and fast motion into two branches of the MLP, enabling fine-grained modeling of high-velocity motion.
> (ii) A local embedding regularizer encourages neighboring Gaussians to move coherently.
> Together, these properties yield locally aligned yet high-frequency trajectories, making E-D3DGS strong in scenes with fast but structured motion.
>
> - **STG (polynomial trajectory model).**
> STG’s polynomial motion parameterization induces globally smooth, low-curvature trajectories, which naturally align with rigid or near-rigid global motion. This makes STG particularly effective in scenes dominated by coherent, large-scale motion patterns.
>
> - **Ex4DGS (interpolation-based deformation).**
> Because Ex4DGS drives motion through keyframe interpolation, Gaussians often follow locally diverse and multi-directional trajectories. This makes Ex4DGS effective in scenes containing irregular or abrupt motion, where interpolation signals introduce heterogeneous displacement patterns.
>
> Taken together, these observations provide a mechanistic explanation for the specialization patterns originally shown in **Fig. 1**. The differences do not arise from noise or dataset artifacts but from how each deformation prior constrains the motion it can express.
>
> We note that these tendencies are representative but not perfectly uniform across all scenes, as real-world sequences often contain mixed or ambiguous motion. Nevertheless, the overall trends consistently explain why no single dynamic GS method performs best across all spatial or temporal regions. This directly motivates the MoE-GS framework, which adaptively routes to the deformation prior most compatible with each region and time step. Additional expert-specific motion statistics and examples are provided in **Appendix F**.
>
> **References:**
>
> [1] Wu, Guanjun, et al. "4d gaussian splatting for real-time dynamic scene rendering." Proceedings of the IEEE/CVF conference on computer vision and pattern recognition. 2024.
>
> [2] Bae, Jeongmin, et al. "Per-gaussian embedding-based deformation for deformable 3d gaussian splatting." European Conference on Computer Vision. Cham: Springer Nature Switzerland, 2024.
>
> [3] Li, Zhan, et al. "Spacetime gaussian feature splatting for real-time dynamic view synthesis." Proceedings of the IEEE/CVF Conference on Computer Vision and Pattern Recognition. 2024.
>
> [4] Lee, Junoh, et al. "Fully explicit dynamic gaussian splatting." Advances in Neural Information Processing Systems 37 (2024): 5384-5409.

---

> > ### Comment · Reviewer_bdci · 2025-11-27
> >
> > I appreciate the authors’ clarification of my questions. However, after carefully reviewing the newly added Appendix E and the multi-view depth consistency results, the weakness 1 mentioned earlier remain largely unresolved.
> >
> > I understand from Appendix E that pixel-level gates can be lifted back to individual Gaussians and used to construct a post-hoc “fused” Gaussian model by reweighting the opacities. However, the other core 3D Gaussian parameters (except opacity) are still disjoint across experts and remain unmodified. The fused representation therefore keeps all expert-specific geometries and only adjusts their visibility, rather than forming a truly unified geometric model. If I misunderstood this point and there actually exists a final, unified Gaussian representation (beyond opacity reweighting), I would appreciate a clarification.
> >
> > The reported depth-consistency improvements then seem to follow mainly from visibility-based selection, rather than from any meaningful merging or reconciliation of the underlying geometries. As a result, the “fused” model still effectively retains multiple, incompatible 3D reconstructions and reduces their visual conflicts by gating some of them out. If the goal is to demonstrate improved geometric reconstruction, a more direct evaluation could involve converting the fused Gaussian field into a mesh or surface representation and measuring geometric consistency explicitly, rather than relying solely on depth consistency derived from rendering.
> >
> > Thus, the rebuttal in fact reinforces my original assessment: the proposed MoE-GS framework primarily improves appearance-level rendering, rather than providing convincing evidence of improved geometric reconstruction or dynamic scene modeling as claimed. If my understanding above is inaccurate, please feel free to refute or clarify it with more precise details.

---

> > > ### Author Response · Authors · 2025-11-30
> > >
> > > Thank you again for carefully reading **Appendix E** and the Multi-view Depth Consistency (MDC) results. In your original review, you wrote:
> > >
> > > > “While I understand that per-Gaussian routing may be technically challenging, I would appreciate clarification on whether the authors believe their Mixture-of-Experts design can genuinely improve geometric reconstruction quality rather than primarily enhancing rendering appearance.”
> > >
> > > We appreciate this focus on geometry and agree that moving from mixtures at the image level toward more unified 3D representations is an important long-term direction. At the same time, we would like to clarify what “reconstruction” typically means in the NeRF / GS setting, and where MoE-GS sits in that landscape.
> > >
> > > ---
> > >
> > > **1. Context: what “reconstruction” usually means in NeRF / GS**
> > >
> > > In both NeRF and Gaussian Splatting work, it is standard practice to treat high-quality **novel view synthesis** as the primary reconstruction objective, while explicit surface or mesh extraction is pursued by a separate line of methods.
> > >
> > > - Classical radiance-field methods such as NeRF [1] and Mip-NeRF360 [2] optimize a volumetric density/radiance field using photometric losses and report PSNR/SSIM on held-out views as the main “3D reconstruction” metric; they do not explicitly output a canonical mesh.
> > > - When explicit geometry is the primary goal (e.g., NeuS-style neural SDFs [3] or SuGaR-style mesh extraction on top of 3DGS [4]), it is handled by dedicated pipelines and evaluated under different metrics and assumptions.
> > >
> > > Within Gaussian Splatting, the original 3DGS paper [5] explicitly states that “our goal is to allow real-time rendering for scenes captured with multiple photos” and only mentions mesh reconstruction as an interesting future direction. Dynamic GS methods such as 4DGS [6], E-D3DGS [7], STG [8], and Ex4DGS [9] similarly target dynamic novel view synthesis and assess geometry implicitly through rendering performance and temporal consistency, without producing a unified mesh.
> > >
> > > MoE-GS intentionally follows this dominant line of work: our aim is to improve the **geometric behavior** (occupancy, visibility, depth coherence) of a dynamic 4D radiance representation under a MoE routing scheme, rather than to deliver a single explicit canonical surface.
> > >
> > > ---
> > >
> > > **2. Scope: geometry-aware routing, not parametric fusion**
> > >
> > > Within this context, we agree that MoE-GS does *not* perform full parametric fusion of expert geometries (i.e., jointly re-optimizing centers, scales, rotations, SH across experts). In that stronger sense of a unified canonical model, we do not claim to solve geometric fusion. Our notion of “geometric reconstruction quality” is more modest and refers to the **consistency** of the volumetric representation—occupancy, visibility, and depth behavior—across views and time.
> > >
> > > Structurally, MoE-GS is a geometry-aware *routing* mechanism across heterogeneous dynamic GS experts, not a full 3D fusion framework. Each expert’s deformation space remains intact; the router decides **which expert’s geometry is trusted where**, instead of collapsing all expert parameters into a single canonical model. The post-hoc “fused” GS in Appendix E is therefore a diagnostic tool: it shows that after routing, the **effective** 3D occupancy (which Gaussians are actually visible along each ray) becomes more coherent across views and time, even though per-expert parameters remain unchanged.
> > >
> > > ---
> > >
> > > **3. What is actually fused: Gaussian-level selection, not just 2D blending**
> > >
> > > It is correct that, in Appendix E, our post-hoc model only reweights opacities. However, in Gaussian Splatting, opacity and transmittance directly govern **which surfaces occlude which and where density actually resides in 3D** along each ray. Changing opacities with fixed centers is therefore not merely a 2D color operation: it changes the effective 3D occupancy and occlusion structure, i.e., which parts of each expert’s geometry are realized in 3D.
> > >
> > > Moreover, although we do not fuse parameters across experts, the learned routing effectively defines a **single fused 4D geometry at the Gaussian level**:
> > >
> > > - for each spatial region and time step, only Gaussians with non-negligible opacity survive;
> > > - Gaussians that are consistently assigned very low opacity can be pruned entirely.

---

> > > ### Author Response · Authors · 2025-11-30
> > >
> > > In other words, MoE-GS is not keeping multiple full 3D reconstructions “alive” and simply averaging their images; it constructs a new scene representation by **selecting which Gaussians from which expert constitute the final 4D geometry**. This is conceptually analogous to the adaptive density control in 3DGS [5], where geometry is shaped not only by continuous parameter updates, but also by creating, refining, or pruning Gaussians based on their contribution to the rendered views. In MoE-GS, the router-driven responsibilities play a similar role across experts: they decide which Gaussians remain active and which are effectively suppressed. It is a different fusion style than parametric averaging, but it operates at exactly the level that is rendered—determining which primitives occupy space and cast occlusions.
> > >
> > > Our MDC metric is computed from rendered depth and is sensitive exactly to these occupancy / ordering changes. If routing were only doing 2D appearance averaging, we would not expect consistent, scene-wide improvements in depth reprojection error across all N3V scenes. Yet MDC improves across all scenes, and spiral-view visualizations (Appendix E and the [project page](https://moegs-moe-gs.static.hf.space/index.html#spiral)) show cleaner depth layering and fewer popping artifacts. We therefore view the Gaussian-level fusion induced by MoE-GS as a meaningful improvement of the underlying 4D occupancy / visibility structure, not only of its 2D appearance.
> > >
> > > We also agree that a mesh- or surface-based evaluation would be a more direct probe of canonical geometry. However, such evaluations typically rely on a different pipeline (e.g., NeuS- or SuGaR-style methods [3, 4]) and additional assumptions or supervision tailored to explicit surface extraction, which go beyond the standard dynamic NVS setting we target here. For MoE-GS, we therefore focused on volumetric metrics such as MDC, which are aligned with NeRF/GS-style radiance representations, and view mesh-space fusion as a natural next step building on the 3D responsibilities exposed
> > > by our method.
> > >
> > > ---
> > >
> > > **4. The role of MoE-GS as a first step**
> > >
> > > At the same time, we see MoE-GS as a **self-contained** contribution within the standard dynamic NVS paradigm. Our diagnostic analysis (Fig. 1) and MoE formulation address a previously under-explored question: how to generalize dynamic GS by adaptively combining experts with different deformation priors. To our knowledge, no prior dynamic GS work has studied this cross-method specialization and routing problem, and our experiments show that doing so yields state-of-the-art dynamic novel view synthesis quality with improved 3D consistency across scenes, regions, and time.
> > >
> > > As you noted in your strengths, our perspective
> > >
> > > > “could motivate subsequent studies toward true 3D-level mixtures or unified dynamic representations.”
> > >
> > > We agree with this and, in the revised conclusion, we explicitly highlight this as a natural but significantly more ambitious future direction. In particular, we point out that the 3D per-Gaussian responsibilities exposed by MoE-GS could serve as a starting point for methods that aim to refine or merge expert geometries more explicitly in 3D. In this sense, we view MoE-GS as a practical and well-scoped first step toward the stronger geometric goals you emphasized, while remaining faithful to the evaluation protocols and objectives that are standard in the NeRF / GS literature today.
> > >
> > > **References:**
> > >
> > > [1] Mildenhall, B., et al. “NeRF: Representing scenes as neural radiance fields for view synthesis.” *Communications of the ACM*, 2021.
> > > [2] Barron, J. T., et al. “Mip-NeRF 360: Unbounded anti-aliased neural radiance fields.” *CVPR*, 2022.
> > > [3] Wang, P., et al. “NeuS: Learning neural implicit surfaces by volume rendering for multi-view reconstruction.” *NeurIPS*, 2021.
> > > [4] Guédon, A., and Lepetit, V. “SuGaR: Surface-aligned Gaussian Splatting for efficient 3D mesh reconstruction and high-quality mesh rendering.” *CVPR*, 2024.
> > > [5] Kerbl, B., et al. “3D Gaussian Splatting for real-time radiance field rendering.” *ACM TOG*, 2023.
> > > [6] Wu, G., et al. “4D Gaussian Splatting for real-time dynamic scene rendering.” *CVPR*, 2024.
> > > [7] Bae, J., et al. “Per-Gaussian embedding-based deformation for deformable 3D Gaussian Splatting.” *ECCV*, 2024.
> > > [8] Li, Z., et al. “Spacetime Gaussian Feature Splatting for real-time dynamic view synthesis.” *CVPR*, 2024.
> > > [9] Lee, J., et al. “Fully explicit dynamic Gaussian Splatting.” *NeurIPS*, 2024.

---

### Official Review · Reviewer_7MH6 · 2025-11-01

**Soundness:** 4
**Presentation:** 4
**Contribution:** 4
**Rating:** 10
**Confidence:** 4

**Summary:**

MoE-GS identifies that different dynamic Gaussian Splatting methods have varying image quality performance across scenes, image patches, and time, and proposes a mixture-of-experts dynamic GS method to adaptively combine the outputs from a few experts, each using a different dynamic Gaussian Splatting method. To stabilize the MoE-GS optimization, the authors propose a novel Volume-aware Pixel Router design to combine expert outputs through differentiable weight splatting, where routing decisions at the Gaussian level are projected to the pixel level to facilitate optimization. To improve efficiency, the authors propose a single-pass multi-expert rendering to reduce GPU kernel launching and memory IO overhead and a gate-aware Gaussian pruning algorithm to remove less-impactful Gaussians in the MoE setting. To further improve efficiency, the authors propose distilling single experts with the help of the MoE model, which they show is superior to training an equivalent single expert from scratch, using only ground-truth images.

**Strengths:**

Very good results supported by comprehensive quantitative experiments (more in the appendix!), images, and an easy-to-read demo webpage.

MoE-GS runs with higher speeds and lower memory requirements (with pruning enabled) compared to only using one of the methods the experts are based on [Table 3], while offering state-of-the-art image quality.

Good applicability/adaptability. Theoretically one can use any dynamic GS method as an "expert" so long as it returns a set of 3D Gaussians at each time step.

The paper supports the need for using different "experts" well, with Figure 1 showing different dynamic GS methods have varying image quality under different situations, thus it makes intuitive sense to combine different methods together.

**Weaknesses:**

Minor weakness: unlike in MoE LLMs where most experts are not executed, the MoE mode here (i.e. without distillation) still runs all experts at all times. The inference speed has potential to be improved further if, we can combine experts with sparsity. Is there a reason this is not done in this paper?

**Questions:**

How are parameters for w_i^{dir} and w_i^{time} initialized? Are they learnable? L281 suggests they're included in the optimization.

I understand that Single-Pass Multi-Expert Rendering improves speed by reducing kernel launches and memory IO, but it also sounds like it will increase peak VRAM, since we could also run the experts sequentially with CPU offloading (probably much slower), or distribute the experts across GPUs (probably almost just as fast)? Have you considered or tested these two alternative strategies?

Figure 1 shows that different dynamic GS methods have differing performance across scenes, image patches, and time. I wonder if MoE-GS can be generalized to have experts that each represent a different dynamic GS method? E.g. having 2 experts, one using 4D GS and one using Ex4DGS?

Minor points:
I couldn't easily find what the abbreviations (Hex., Poly., Per., inter.) stand for in Figure 1. I know that this is explained in the main text, but it'd be nice to include what these are in the captions. Also, why is "inter" not capitalized while the other abbreviations are?

Variables in Section 3.2 are defined and used confusingly. How do the "2D"-subscripted variables in eq. (7) relate to eq. (6)? w_i is defined in eq. (6) but never used or explained. I'm assuming w_{2D} is a collection of w_i's, but this is not explained.

---

> ### Author Response · Authors · 2025-11-21
> **Comment 1/2 for Reviewer 7MH6**
>
> We thank the reviewer for the careful reading and thoughtful comments. Below, we address the reviewer’s questions and provide detailed explanations. The corresponding revisions in the manuscript are marked in blue for clarity. We hope that the responses below, along with the updated manuscript, address the reviewer’s concerns thoroughly. Due to character limits on OpenReview, our response has been divided into two parts for ease of readability. Thank you for your understanding.
>
> **Weakness 1:** Minor weakness: unlike in MoE LLMs where most experts are not executed, the MoE mode here (i.e. without distillation) still runs all experts at all times. The inference speed has potential to be improved further if, we can combine experts with sparsity. Is there a reason this is not done in this paper?
>
> **Response:** Thank you for raising this valuable point. We agree that sparse expert activation—similar to MoE usage in LLMs—could further improve inference efficiency. In our setting, however, we intentionally avoid sparsity during training for a structural reason:
>
> Dynamic GS experts operate in heterogeneous and non-alignable 3D deformation spaces (canonical HexPlane, per-Gaussian volumetric deformation, polynomial trajectory, and keyframe interpolation). Since these representations are fundamentally different and do not share a common latent space, skipping experts early causes unstable geometry selection, inconsistent depth/visibility ordering, and unreliable gradient signals. Executing all experts ensures that the router learns a stable and geometry-aware selection field across these diverse priors.
>
> That said, we fully agree that sparse expert activation is a promising extension. Once stable routing is learned, sparsity (e.g., top-2 selection or thresholding based on expert weights or 3D responsibilities) can be applied post-training to accelerate inference. We have included this direction as a natural future extension.
>
> **References:**
>
> [1] Wu, Guanjun, et al. "4d gaussian splatting for real-time dynamic scene rendering." Proceedings of the IEEE/CVF conference on computer vision and pattern recognition. 2024.
>
> [2] Bae, Jeongmin, et al. "Per-gaussian embedding-based deformation for deformable 3d gaussian splatting." European Conference on Computer Vision. Cham: Springer Nature Switzerland, 2024.
>
> [3] Li, Zhan, et al. "Spacetime gaussian feature splatting for real-time dynamic view synthesis." Proceedings of the IEEE/CVF Conference on Computer Vision and Pattern Recognition. 2024.
>
> [4] Lee, Junoh, et al. "Fully explicit dynamic gaussian splatting." Advances in Neural Information Processing Systems 37 (2024): 5384-5409.
>
> ---
>
> **Question 1:** How are parameters for w_i^{dir} and w_i^{time} initialized? Are they learnable? L281 suggests they're included in the optimization.
>
> **Response:**
> Both $w_i^{dir}$ and $w_i^{time}$ are learnable parameters, and they are jointly optimized with all other components of MoE-GS. To avoid introducing any handcrafted directional or temporal bias, we initialize both parameters using a neutral near-zero initialization. This allows the router to begin from an unbiased state and to learn meaningful directional and temporal sensitivities directly from data.
>
> We have added this clarification to the manuscript in **Appendix C.3 (Stage 2: Router Training)**.
> The updated text now explicitly states both the learnability and initialization strategy of these parameters.
>
> ---
>
> **Question 2:** I understand that Single-Pass Multi-Expert Rendering improves speed by reducing kernel launches and memory IO, but it also sounds like it will increase peak VRAM, since we could also run the experts sequentially with CPU offloading (probably much slower), or distribute the experts across GPUs (probably almost just as fast)? Have you considered or tested these two alternative strategies?
>
>
> **Response:** Thank you for the insightful question. We explored both alternatives—sequential expert execution with CPU offloading and distributing experts across multiple GPUs—but we found that each approach presents practical limitations in dynamic Gaussian Splatting.
>
> - **Sequential execution** becomes prohibitively slow due to repeated PCIe transfers. Dynamic GS requires high-frequency random access to Gaussian attributes during splatting, and the latency introduced by host–device transfers prevents effective kernel fusion.
>
> - **Multi-GPU distribution** offers limited benefit because each expert resides in a heterogeneous 3D deformation space. Distributing experts requires duplicating Gaussian buffers and synchronizing per-pixel accumulations across GPUs, introducing substantial inter-GPU communication with little net speedup.
>
> For these reasons, we found that Single-Pass Multi-Expert Rendering provides the most stable and efficient trade-off for MoE-GS workloads. A detailed clarification of these findings has been added to **Appendix C.4** in the revised manuscript.
>
> ---

---

> > ### Author Response · Authors · 2025-11-21
> > **Comment 2/2 for Reviewer 7MH6**
> >
> > **Question 3:** Figure 1 shows that different dynamic GS methods have differing performance across scenes, image patches, and time. I wonder if MoE-GS can be generalized to have experts that each represent a different dynamic GS method? E.g. having 2 experts, one using 4D GS and one using Ex4DGS?
> >
> > **Response:** Thank you for the thoughtful question. Yes—MoE-GS is designed to generalize across heterogeneous dynamic GS methods, and using different 4D representations as experts (e.g., 4DGS and Ex4DGS) is fully supported by our framework.
> >
> > MoE-GS does not assume a shared latent deformation space across experts.
> > Each expert operates in its own 3D deformation representation (e.g., canonical HexPlane for 4DGS, keyframe interpolation for Ex4DGS, per-Gaussian deformation for E-D3DGS, etc.), and the router learns to select the expert whose deformation prior best explains the local motion and spatial structure.
> >
> > While our main experiments use a fixed heterogeneous expert set for clarity, we verified that MoE-GS can successfully mix experts drawn from different dynamic GS families due to its deformation-agnostic router design. We have explicitly clarified this capability—and listed the exact expert configurations used—in **Section 4.1 (Experimental Setup)** of the revised manuscript.
> >
> > **References:**
> >
> > [1] Wu, Guanjun, et al. "4d gaussian splatting for real-time dynamic scene rendering." Proceedings of the IEEE/CVF conference on computer vision and pattern recognition. 2024.
> >
> > [2] Lee, Junoh, et al. "Fully explicit dynamic gaussian splatting." Advances in Neural Information Processing Systems 37 (2024): 5384-5409.
> >
> > [2] Bae, Jeongmin, et al. "Per-gaussian embedding-based deformation for deformable 3d gaussian splatting." European Conference on Computer Vision. Cham: Springer Nature Switzerland, 2024.
> >
> > ---
> >
> >
> >
> > **Question 4:** Minor points: I couldn't easily find what the abbreviations (Hex., Poly., Per., inter.) stand for in Figure 1. I know that this is explained in the main text, but it'd be nice to include what these are in the captions. Also, why is "inter" not capitalized while the other abbreviations are?
> >
> > **Response:** Thank you for pointing this out. We agree that the abbreviations in **Figure 1** were not immediately clear. In the revised manuscript, we have added the full names of all deformation types (HexPlane, Polynomial Trajectory, Per-Gaussian Deformation, and Interpolation-based Deformation) directly in the figure caption for clarity. We have also standardized the capitalization and changed “inter” to “Inter.” to ensure consistency with the other abbreviations.
> >
> > ---
> >
> > **Question 5:** Variables in Section 3.2 are defined and used confusingly. How do the "2D"-subscripted variables in eq. (7) relate to eq. (6)? w_i is defined in eq. (6) but never used or explained. I'm assuming w_{2D} is a collection of w_i's, but this is not explained.
> >
> > **Response:** Thank you for pointing out the inconsistency in the notation in **Section 3.2**. We agree that the relationship between the variables in Eq. (6) and the 2D-subscripted variables in Eq. (7) was not clearly explained.
> >
> > In the revised manuscript, we have clarified the following:
> >
> > - **$w_i$** in Eq. (6) denotes the *per-Gaussian weight* for the $i$-th Gaussian in 3D.
> > - These volumetric weights are projected to the image plane through the differentiable Gaussian rasterizer, producing pixel-aligned embeddings $w_{2D}(u)$ in Eq. (7).
> >
> > - Accordingly, $w_{2D}(u)$ represents the aggregated 2D projection of all relevant $w_i$ that contribute to pixel $u$.
> >
> > - Eq. (7) then applies the shared MLP to this rasterized 2D feature to compute the router logits $R’_k(u)$.
> >
> > We have updated **Section 3.2** and the caption of Eq. (7) to explicitly state this relationship and remove the ambiguity.
> > Thank you again for raising this point — it has helped improve the clarity of our notation.

---

> > > ### Comment · Reviewer_7MH6 · 2025-11-21
> > >
> > > I appreciate the clarification of my (and other reviewers') questions from the authors and especially the additional ablations/experiments added to the revised manuscript. My questions have been answered.
> > >
> > > I believe that the paper is sufficiently complete under the scope it positions itself in: to achieve better-quality novel view synthesis by combining different dynamic GS methods which have different inductive biases in how they model motion. However, as echoed by Reviewer AHQB and Reviewer eMnv, since the paper's method is named after "Mixture of Experts," it is easy to expect that the paper offers a way to reduce the training and inference-time FLOPs at the same NVS quality, but that is not what the paper offers. The FLOPs and VRAM usage scale linearly w.r.t. the number of experts. I believe that this is a good paper, but it can benefit from stating clearly what the method aims to do, and what it does not. It is very easy for readers to expect training and inference efficiency gains (in FLOPs or wall-clock-time, quality-equalized) once a paper mentions "Mixture of Experts," but as far as I'm aware that is not the aim.

---

> > > > ### Author Response · Authors · 2025-11-21
> > > >
> > > > Thank you for the helpful clarification and for highlighting this potential source of confusion for readers. Following your suggestion (and the related concerns raised by Reviewers AHQB and eMnv), we have explicitly clarified the scope of MoE-GS in both the abstract and the introduction.
> > > >
> > > > In particular, we now state that, unlike sparsity-oriented MoE architectures in large language models, MoE-GS is **not** intended to reduce training or inference-time FLOPs or memory usage. Instead, its primary goal is to improve dynamic novel view synthesis and reconstruction quality by combining heterogeneous deformation priors, while our efficiency components (single-pass multi-expert rendering, pruning, and distillation) are complementary mechanisms to keep this additional cost practical. We hope this clarification makes the intended purpose of our MoE design clear to readers.

---

> > > > > ### Comment · Reviewer_7MH6 · 2025-11-21
> > > > >
> > > > > My apologies. I went back and checked the updated blue parts in the abstract and intro on page 2.
> > > > > The abstract now includes
> > > > > > Unlike sparsity-oriented MoE architectures in large language models, MoE-GS is designed to improve dynamic novel
> > > > > view synthesis quality by combining heterogeneous deformation priors, rather than
> > > > > to reduce training or inference-time FLOPs.
> > > > >
> > > > > While the intro at the bottom of page 2 now includes
> > > > >
> > > > > > It is important to clarify that, unlike sparsity-driven
> > > > > MoE architectures in large language models, MoE-GS is not intended to reduce FLOPs or memory
> > > > > usage. Our primary goal is to increase representational capacity and improve dynamic reconstruction
> > > > > quality by combining heterogeneous deformation priors; the efficiency techniques we introduce are
> > > > > complementary mechanisms to keep this additional cost practical.
> > > > >
> > > > > These clarifications addressed my concerns.

---

### Author Response · Authors · 2025-11-21
**Summary of Revisions**

## **Summary of Revisions**

We thank all reviewers for their insightful and constructive feedback. Their comments greatly improved the clarity, rigor, and completeness of the manuscript. Below is a summary of the key updates. Additional visualizations and videos are provided on our **[project page](https://huggingface.co/spaces/moegs/MoE-GS)**.

---

## **Key Updates**

### **Main Paper Revisions**

- **Clarified Motivation for MoE-GS**
  Expanded **Sec. 1** to clearly explain why heterogeneous deformation priors in dynamic GS naturally motivate a mixture-of-experts framework. (bdci , AHQB)

- **Geometry- and Gaussian-Level Interpretation Added**
  Added a brief explanation in **Sec. 3.2** showing how pixel router weights can be lifted back to per-Gaussian responsibilities, enabling simple post-hoc 3D fusion. (bdci)

- **Training-Time Analysis Added**
  Added a detailed study in **Sec. 4.2** showing that MoE-GS does not require fully converged experts.
  Even with 20% training budget per expert, MoE-GS still outperforms fully trained single experts. (AHQB, eMnv)

---

### **Appendix Updates**

- **Appendix C — Router & Rendering Implementation Details**
  Added initialization details for learnable directional/temporal weights ($w_i^{dir}$, $w_i^{time}$) and provided a detailed comparison of rendering strategies, explaining why Single-Pass Multi-Expert Rendering is adopted over sequential execution or multi-GPU distribution. (7MH6)

- **Appendix E — Gaussian-Level Interpretation & Fusion**
 Added derivations showing how pixel-level router outputs lift back to per-Gaussian volumetric responsibilities and how this enables post-hoc fused Gaussian models. Also included Multi-view Depth Consistency results and fused-Gaussian visualizations demonstrating improved 3D occupancy coherence. Corresponding qualitative results and videos are available on the **[project page](https://moegs-moe-gs.static.hf.space/index.html#spiral)** (bdci, AHQB, eMnv)

- **Appendix F — Motion Behavior Analysis**
  Added a causal analysis explaining why each expert specializes in different motion regimes (fast, irregular, smooth, static) based on their deformation priors, with corresponding trajectory visualizations provided on the **[project page](https://moegs-moe-gs.static.hf.space/index.html#analysis)** (bdci, AHQB, eMnv)

- **Appendix G — Multi-Run Stability**
  Added a 5× retraining evaluation, showing that MoE-GS achieves higher median performance and lower variance across repeated trainings. (eMnv)

- **Appendix H — Distillation Weighting Ablation**
  Compared different weighting strategies (w/o weighting vs. routing-weight weighting) and showed that routing-based weights yield the strongest improvements. (eMnv)

---

### **Other Improvements**

- **Figure Caption Fixes**
  **Fig. 1** now clarifies all abbreviations (HexPlane, Polynomial, Per-Gaussian, Interpolation) and uses consistent capitalization. (7MH6)

- **Improved Method Explanation**
  Updated **Sec. 3.2** to clearly define how 3D per-Gaussian weights are projected into rasterized 2D features, and to explain the initialization and learnability of directional and temporal weighting parameters. (7MH6)

- **Expert Configurations Clarified**
  **Sec. 4.1** now explicitly lists expert sets for N = 2, 3, 4. (7MH6)

---

## **Key Contributions Strengthened**

- MoE-GS leverages complementary deformation priors to produce a globally coherent 4D representation that no individual expert can achieve.
- The fused-Gaussian model shows clear geometric benefits in Multi-view Depth Consistency, not just improvements in PSNR/SSIM.
- Added extensive ablations confirming the robustness, stability, and practical efficiency of the framework.

---

We believe these revisions address all reviewer concerns and substantially strengthen the clarity and technical contribution of the paper.
**Thank you again for your constructive feedback!**

---

### Author Response · Authors · 2025-12-01
**Final AC Summary (3/3)**

# 2. Summary of Minor Revisions

These issues were detailed but not conceptual, and were all fully addressed.

---
### **A. Causal Explanation of Expert Specialization (Fig. 1)**
*(Raised by: bdci, AHQB, eMnv)*
- Introduced a **causal motion-behavior analysis** in **Appendix F**
- Added a **motion-trajectory diagnostic** on the **[project page](https://moegs-moe-gs.static.hf.space/#analysis)**
- Updated **Fig. 1** to reflect expert-specific specialization patterns
- Provided mechanistic explanations rooted in each method’s deformation prior

→ **All concerns resolved.**

---

### **B. Training Cost & Reduced-Budget Settings**
*(Raised by: AHQB, eMnv)*
- Reported per-expert training times in **Sec. 4.2**
- Added **20% budget experiment**, showing MoE-GS surpasses fully trained single experts even with reduced expert training
- Clarified that experts need not be fully converged

→ **All concerns resolved.**

---

### **C. Router Initialization, Temporal Weighting, and Eq. (6) Clarity**
*(Raised by: 7MH6, AHQB)*
- Full initialization procedure added in **Appendix C.3**
- Clarified temporal weighting in **Eq. (6)**
- Expanded notation and descriptions

→ **All concerns resolved.**

---

### **D. Missing or Inconsistent Notation**
*(Raised by: 7MH6, AHQB)*
- Standardized abbreviations and symbols across **Fig. 1** and **Sec. 3.2**
- Added missing definitions throughout the method section

→ **All concerns resolved.**

---

### **E. Memory Usage & Rendering Strategy**
*(Raised by: 7MH6)*
- Added **Appendix C.4** comparing sequential, single-pass, and multi-GPU execution strategies
- Justified why single-pass multi-expert rendering is optimal in our setting

→ **All concerns resolved.**

---

### **F. Expert Combinations & Model Configuration Clarity**
*(Raised by: 7MH6)*
- Listed all expert combinations for N=2–4 in **Sec. 4.1**
- Clarified heterogeneous expert mixing and routing stability

→ **All concerns resolved.**

---

### **G. Additional Ablations & Stability Studies**
*(Raised by: eMnv)*
- Added **geometry-level MDC analysis** (Appendix E)
- Added **Expert motion–behavior analysis** (Appendix F)
- Added **multi-run stability** results (Appendix G)
- Added **distillation weighting–strategy ablation** (Appendix H)

→ **All concerns resolved.**

---

### **H. Inclusion of MoE-GS Results in Figure 1**
*(Raised by: AHQB)*
- Updated **Figure 1** to include MoE-GS performance, as requested
- Ensured consistent comparison across all experts

→ **All concerns resolved.**

---

### **I. Explanation of HyperReel Exclusion**
*(Raised by: AHQB)*
- Clarified that HyperReel uses a radiance-field representation incompatible with Gaussian-based deformation
- Explained why it cannot be incorporated as an MoE expert

→ **All concerns resolved.**

---

### **J. Distillation hyperparameter in Eq. (11)**
*(Raised by: eMnv)*
- Clarified the role of α in balancing MoE-guided distillation with ground-truth supervision
- Explained why α is fixed to a balanced default in all experiments

→ **All concerns resolved.**

---

# 3. Summary of reviewer landscape

In summary, three reviewers **(7MH6: 10, AHQB: 6, eMnv: 6)** were fully satisfied under the clarified scope and explicitly maintained positive recommendations, **emphasizing the novelty and effectiveness of MoE-GS for dynamic NVS** and improved 4D consistency. Reviewer **bdci (2)** remains critical primarily on conceptual grounds, asking for canonical 3D fusion and mesh-level geometry—objectives outside the standard dynamic GS / radiance-field NVS setting. The discussion ended prematurely due to the OpenReview incident, and we hope the added diagnostics and clarified scope address the conceptual concerns within the intended setting.

---

### **References**

[1] Wu, G., et al. “4D Gaussian Splatting for real-time dynamic scene rendering.” *CVPR*, 2024.

[2] Bae, J., et al. “Per-Gaussian embedding-based deformation for deformable 3D Gaussian Splatting.” *ECCV*, 2024.

[3] Li, Z., et al. “Spacetime Gaussian Feature Splatting for real-time dynamic view synthesis.” *CVPR*, 2024.

[4] Lee, J., et al. “Fully explicit dynamic Gaussian Splatting.” *NeurIPS*, 2024.

[5] Oksuz, K., et al. "Mocae: Mixture of calibrated experts significantly improves object detection." *TMLR*, 2024.

[6] Kuzucu, S., et al. "On calibration of object detectors: Pitfalls, evaluation and baselines." *ECCV*, 2024

[7] Mildenhall, B., et al. “NeRF: Representing scenes as neural radiance fields for view synthesis.” *CACM*, 2021.

[8] Kerbl, B., et al. “3D Gaussian Splatting for real-time radiance field rendering.” *ACM TOG*, 2023.

[9] Wang, P., et al. "Neus: Learning neural implicit surfaces by volume rendering for multi-view reconstruction." *NeurIPS*, 2021.

[10] Guédon, A., et al. “SuGaR: Surface-aligned Gaussian Splatting for efficient 3D mesh reconstruction and high-quality mesh rendering.” *CVPR*, 2024.

---

---

### Author Response · Authors · 2025-12-01
**Final AC Summary (2/3)**

### **B. Scope of “reconstruction” in dynamic GS**
*(Raised by: bdci)*

Reviewer raised a conceptual objection about what “reconstruction” should mean:

> “In 3D reconstruction, novel view synthesis is merely a means to validate geometry,
>  not the goal itself.”

From this perspective, they expected MoE-GS to perform **canonical 3D geometry fusion**—e.g., unifying Gaussian centers, scales, rotations, and SH parameters into a single, explicit surface representation.

In contrast, the prevailing **problem setting** in NeRF / GS [7, 8] literature is radiance-field–based **novel view synthesis (NVS)**. The primary objective in this line of work is to learn a volumetric radiance or Gaussian field that produces high-fidelity novel views, while explicit mesh or canonical surface reconstruction is treated as a *separate* research track with its own methods and evaluation metrics.

In that radiance-field setting, good 3D structure is certainly desirable, but it is pursued insofar as it supports stable and consistent NVS; explicit canonical surfaces or meshes are typically addressed by dedicated pipelines such as NeuS [9] or SuGaR [10].

Foundational NeRF and GS works optimize volumetric radiance or Gaussian fields for high-fidelity NVS and treat explicit mesh extraction as future or separate work.
For example, the original 3DGS paper [8] states:

> “Our goal is to allow real-time rendering for scenes captured with multiple photos.” (Introduction Sec.)
>
> “It would be interesting to see if our Gaussians can be used to perform mesh
>  reconstructions of the captured scene.” (Discussion and Conclusions Sec.)

Dynamic GS methods—including 4DGS [1], E-D3DGS [2], STG [3] and Ex4DGS [4]—explicitly adopt this NVS formulation: they target dynamic view synthesis, and geometric behavior is evaluated implicitly via cross-view and temporal consistency rather than mesh-level fusion.
As STG [8] puts it:

> “We present a novel representation based on Spacetime Gaussians for dynamic
>  view synthesis.” (Conclusions Sec.)

MoE-GS follows this radiance-field paradigm. Rather than aiming to output a single explicit canonical surface, MoE-GS is designed to **improve the geometric behavior of the underlying 4D volumetric field**—its occupancy, visibility, and depth coherence— through geometry-aware routing across heterogeneous deformation priors, while achieving state-of-the-art dynamic NVS performance.

In direct response to Reviewer bdci’s concern about geometry, we added **Appendix E** to make the effect of routing on 3D occupancy and visibility explicit, rather than relying only on PSNR/SSIM. Appendix E introduces a **post-hoc fused model**, rendered with the standard 3DGS renderer. This model is not an additional optimization stage; it is purely a diagnostic tool that:

- lifts per-pixel router outputs back to 3D via per-Gaussian responsibilities, and
- reweights only each Gaussian’s opacity accordingly.

Although centers and scales remain unchanged, modifying opacity directly alters which Gaussians occupy space, how occlusions are ordered, and how surfaces emerge along each ray—i.e., the **effective 4D geometry**. The fused model exhibits:

- more coherent 3D occupancy,
- improved depth layering, and
- fewer conflicting structures across views and time.

Together with the consistent Multi-view Depth Consistency (MDC) improvements across all N3V scenes, these results show that MoE-GS improves geometric behavior through routing alone, even without performing parametric geometry fusion. Spiral-view visualizations (**[project page](https://moegs-moe-gs.static.hf.space/#spiral)**) further reveal a coherent 3D occupancy structure emerging from the mixture.

We also agree that a mesh- or surface-based evaluation would provide a more direct probe of canonical geometry. However, such evaluations typically rely on a different class of methods (e.g., NeuS-/SuGaR-style pipelines [9, 10]) and additional assumptions tailored to explicit surface extraction, which go beyond the standard dynamic NVS setting that current dynamic GS methods (including ours) target. For MoE-GS, we therefore focused on radiance-field–aligned volumetric metrics such as MDC, and view mesh-space fusion as a natural next step building on the 3D responsibilities exposed by our method.

Because this post-hoc fused model uses exactly the same Gaussian parameterization and renderer as standard 3DGS, it is also directly compatible with existing GS-based mesh extraction pipelines (e.g., SuGaR [10]). We view such canonical fusion or mesh reconstruction on top of MoE-GS as a natural but substantially more ambitious **future work**, rather than as part of the current paper’s radiance-field NVS scope.

→ **Scope clarification, diagnostic 3D analysis, and incorporated the reviewer’s suggestion as future-work direction; all concerns addressed.**

---

---

### Author Response · Authors · 2025-12-01
**Final AC Summary (1/3)**

**To the Area Chair,**

We sincerely appreciate the additional burden placed on ACs in this cycle due to the OpenReview incident. Since reviewers can no longer revise scores or continue the discussion, we have prepared this summary to make your evaluation easier and to present the technical points in the **correct scope**, as established in our rebuttal and in the dynamic GS / NVS literature. We also encourage you to refer to the **[project page](https://moegs-moe-gs.static.hf.space/)**, where comprehensive results and diagnostics are provided.

For context, we briefly summarize the core technical contributions of our submission. Concretely, MoE-GS (i) introduces a **Mixture-of-Experts framework over heterogeneous dynamic GS methods** (4DGS [1], E-D3DGS [2], STG [3], Ex4DGS [4]) with a Volume-aware Pixel Router that computes per-pixel expert weights from 3D Gaussian features via differentiable weight splatting; (ii) demonstrates **state-of-the-art dynamic novel view synthesis** on the N3V and Technicolor datasets, while improving 4D occupancy/visibility consistency as measured by our Multi-view Depth Consistency (MDC) metric and spiral-view diagnostics; and (iii) proposes **practical efficiency mechanisms** (single-pass multi-expert rendering, gate-aware pruning, and MoE-guided distillation) that make the multi-expert design usable in practice without modifying the expert architectures.

Across all four reviews, two conceptual themes appear:

**1. The role of MoE-GS relative to sparsity-oriented MoE in LLMs (Reviewers 7MH6, AHQB, eMnv).**

**2. The meaning of “reconstruction” in dynamic Gaussian Splatting (Reviewer bdci).**

Because many reviewer-specific questions are rooted in these two themes, we treat them as **Major Revisions** and clarify our intended scope below. All remaining reviewer requests—regarding training cost, initialization, ablations and notation—are summarized in the Summary of **Minor Revisions** that follows.

---

# 1. Summary of Major Revisions (Conceptual Scope Clarifications)
These were the two conceptual themes underlying many reviewer questions.


### **A. Scope of Mixture-of-Experts (efficiency vs. routing)**
*(Raised by: 7MH6, AHQB, eMnv)*

A recurring question was whether MoE-GS should resemble sparsity-driven LLM MoEs, i.e., using sparse expert activation to reduce FLOPs.

In our rebuttal, we clarified that:

- MoE-GS is designed to **route across distinct 3D deformation spaces** in order to generalize across multiple, co-existing motion regimes in real scenes—not to provide sparsity.
- Each dynamic GS expert (4DGS [1], E-D3DGS [2], STG [3], Ex4DGS [4]) is structurally specialized for different motion patterns (fast, irregular, locally coherent, or smooth global trajectories), due to its own deformation prior.
- This specialization is structural rather than noise-driven. We added **Appendix F** to provide a causal explanation and motion-behavior analysis for each expert, supporting the MoE design.

Notably, recent vision MoE research in object detection—such as Mixture of Calibrated Experts (MOCAE) [5] and post-hoc detector calibration frameworks [6] shows that combining heterogeneous expert models and properly calibrating their predictions can yield accuracy superior to any single expert, without relying on computational sparsity. In a similar spirit, MoE-GS uses routing across specialized dynamic GS experts to robustly cover diverse motion regimes that no single 4DGS model handles uniformly well.

To avoid ambiguity for readers, we explicitly incorporated this into the revised **Abstract** and **Introduction**, which now states:

> “Unlike sparsity-oriented MoE architectures in large language models, MoE-GS is
> designed to improve dynamic novel view synthesis quality by combining heterogeneous
> deformation priors, rather than to reduce training or inference-time FLOPs.” (Abstract Sec.)

> “It is important to clarify that, unlike sparsity-driven MoE architectures in large language models, MoE-GS is not intended to reduce FLOPs or memory usage.” (Introduction Sec.)

→ **All concerns resolved.**

---

---

### Meta-Review · Area_Chair_wZt1 · 2026-01-09

**Summary:**

This paper proposes MoE-GS, a framework to integrate Mixture-of-Experts (MoE) techniques into dynamic 3D Gaussian Splatting. The paper addresses limitation of not having an optimal defomation prior in the dynamic GS problem by mixing different expert GS models. The authors propose a Volume-aware Pixel Router that combines the efficiency of pixel-level router with the flexibility of per-GS router. Performance optimization techniques such single-pass multi-expert rendering is also introduced to make the rendering faster.

The paper initially received polarized reviews (10, 2, 6, 6), with the major contention on whether the MoE method introduce can effectively improve the geometric reconstruction. The authors hence make an extension of the proposed method to perform post-hoc geometry fusion of the expert models, which interestingly also shows improvement in geometric quality. While the debate over the goal of 3DGS remains a philosophical divide in the field, the authors' addition of the MDC metric and responsibility-weighted lifting effectively bridges this gap. Several main concerns are addressed adequately in the rebuttal. I therefore recommend acceptance.

**Reviewer Concerns:**

The following concerns are addressed in rebuttal:

-  Conceptual Framing of MoE. Several reviewers (AHQB, eMnv) initially noted a mismatch between this approach and MoEs in Large Language Models (LLMs), aiming to get better performance with smaller compute cost. The authors clarified that the proposed MoE-GS can also surpass single expert's quality. Both reviewers noted the response is satisfactory.

- Geometric vs. Appearance Improvement. The most significant point of contention came from Reviewer bdci, who argued that the framework performs image-level ensembling, which does not translate to better geometric reconstruction. The authors introduced Multi-view Depth Consistency (MDC) as a metric, proving that MoE-GS improves underlying 3D stability, not just RGB appearance. They also detailed a "lifting" procedure in Appendix E that maps pixel gates back to 3D Gaussian responsibilities. The author demonstrated that quality also improves in geometry-sensitive metrics like MDC.

- Training Efficiency. Reviewer eMnv expressed concern regarding the cost of training multiple baseline models. The authors provided an ablation showing that MoE-GS outperforms fully-trained single-expert baselines even when its own experts are trained with only 20% of their standard budget. So it can maintain a reasonable training cost. The reviewer seems satisfied by this response.

The following concerns remain outstanding:

- Debate on whether the scene modeling is improved: Even after the authors provided Multi-view Depth Consistency (MDC) results and a "post-hoc fusion" method in Appendix E, the reviewer remained unconvinced. They noted that simply reweighting opacities still retains "multiple, incompatible 3D reconstructions" and merely gates out visual conflicts rather than performing meaningful geometric merging. The authors present a counterargument to the prevailing view of the true "goal" of reconstruction in 3DGS/Nerf. While this is an interesting debate that may not have an immediate verdict, it will not likely undermine the quality of this work.

**Reviewer Scores:**

Reviewer 7MH6 (initially Strong Accept): This reviewer provided the highest score, praising the strong results supported by comprehensive experiments and the intuitive need for combining diverse dynamic GS methods. They will likely maintain the score.

Reviewer AHQB & eMnv (initially Marginal Accept): Both reviewers were satisfied with the rebuttals regarding training efficiency and the necessity of the MoE structure for handling co-existing motion regimes. They acknowledged the framework as a novel way to ensemble various 4DGS methods.

Reviewer bdci (initially Reject): This reviewer maintained a strong negative stance, arguing that the method is a conceptual mismatch for 3D reconstruction. The debate between the reviewer and the authors over whether the 3DGS, or specifically the proposed MoE-GS, has achieved true reconstruction will likely not be resolved in a short time, as it is more philosophical than technical.

---

### Decision · Program_Chairs · 2026-01-26

Accept (Poster)